# Exon junction complex proteins bind nascent transcripts independently of pre-mRNA splicing in *Drosophila melanogaster*

**Subhendu Roy Choudhury[1†], Anand K Singh[1†], Tina McLeod[1], Marco Blanchette[2], Boyun Jang[3‡], Paul Badenhorst[3], Aditi Kanhere[1], Saverio Brogna[1*]**

[1]School of Biosciences, University of Birmingham, Birmingham, United Kingdom; [2]Stowers Institute for Medical Research, Kansas city, United States; [3]Institute of Biomedical Research, University of Birmingham, Birmingham, United Kingdom

**Abstract** Although it is currently understood that the exon junction complex (EJC) is recruited on spliced mRNA by a specific interaction between its central protein, eIF4AIII, and splicing factor CWC22, we found that eIF4AIII and the other EJC core proteins Y14 and MAGO bind the nascent transcripts of not only intron-containing but also intronless genes on *Drosophila* polytene chromosomes. Additionally, Y14 ChIP-seq demonstrates that association with transcribed genes is also splicing-independent in *Drosophila* S2 cells. The association of the EJC proteins with nascent transcripts does not require CWC22 and that of Y14 and MAGO is independent of eIF4AIII. We also show that eIF4AIII associates with both polysomal and monosomal RNA in S2 cell extracts, whereas Y14 and MAGO fractionate separately. Cumulatively, our data indicate a global role of eIF4AIII in gene expression, which would be independent of Y14 and MAGO, splicing, and of the EJC, as currently understood.

*For correspondence: s.brogna@ bham.ac.uk

†These authors contributed equally to this work

Present address: ‡Asan Medical Center, Seoul, South Korea

Competing interests: The authors declare that no competing interests exist.

## Introduction

Pre-mRNA splicing is typically co-transcriptional, yet by modulating the protein composition and conformation of the ribonucleoprotein complex consisting of mature mRNA and associated proteins (mRNP), it can affect cytoplasmic processes such as mRNA localization, translation and nonsense-mediated mRNA decay (NMD) (*Dreyfuss et al., 2002*; *Le Hir et al., 2016*; *Moore and Proudfoot, 2009*). The main distinctive feature of spliced mRNAs appears to be the presence of the exon junction complex (EJC), a four-protein core-complex consisting of eIF4AIII, Y14, MAGO and MLN51, which is typically deposited 20–24 nucleotides (nt) upstream of the exon junctions in mammalian cells (*Kataoka et al., 2001*; *Le Hir et al., 2000*; *Saulière et al., 2012*; *Singh et al., 2012*). Such sequence-independent but position-specific deposition is initiated by the recruitment of the EJC core protein eIF4AIII to the spliceosome via its association with spliceosome component CWC22 (*Alexandrov et al., 2012*; *Barbosa et al., 2012*; *Steckelberg et al., 2012*). While associated with CWC22, eIF4AIII is in an inactive conformation which is incompatible with ATP and RNA binding, and therefore, with EJC assembly (*Buchwald et al., 2013*). The current model predicts that following exon ligation, eIF4AIII dissociates from CWC22, binds the spliced mRNA, typically at the −24 nt position of exon junction, and then triggers EJC assembly by recruiting the three remaining EJC core proteins, Y14, MAGO and MLN51 (*Ballut et al., 2005*; *Bono et al., 2006*; *Steckelberg et al., 2015*). The association of MAGO and Y14 stabilizes binding of the complex to mRNA by inhibiting eIF4AIII ATPase activity (*Ballut et al., 2005*).

**eLife digest** Cells and organisms survive and thrive in large part due to the activities of thousands of proteins. The instructions for making these proteins are found in the DNA sequences of genes. However, these genes also tend to contain large sections called introns that do not encode protein.

To make a protein, the gene's full sequence is first copied to a temporary molecule called pre-messenger RNA (pre-mRNA for short). The introns are then removed from the pre-mRNA in a process known as splicing in the cell nucleus, during which the remaining regions of the molecule, called exons, are joined together to form a mature mRNA molecule. This mature mRNA can then move out of the cell nucleus and be used as a template to build proteins around the cell.

Intriguingly, splicing of the pre-mRNAs in the nucleus affects how the mRNA is used to make proteins in the cytoplasm of the cell. This nucleus-cytoplasm connection is currently explained by the so-called exon junction complex, which is thought to attach to mature mRNAs at the junction between two exons and stay bound until the mRNA moves to the cytoplasm. Evidence suggests the exon junction complex affects how the mRNA is used to make protein, yet little is known about how it would do so.

Choudhury, Singh et al. examined how exon junction complex proteins bind to newly made RNA in salivary gland cells of fruit flies and in cultured cells. Contrary to expectations, the three proteins thought to make the central part of the exon junction complex were found on different mRNAs and regardless of whether the mRNAs derived from genes with introns. Specifically, one of these proteins – eIF4AIII – can remain on the mRNA independently of the two other exon junction complex proteins or CWC22, a protein required for splicing. CWC22 is also thought to be required for the complex to be deposited precisely at exon junctions in human cells.

Overall, it appears that our current understanding of the exon junction complex needs to be revised. The findings presented by Choudhury, Singh et al. predict alternative roles for these proteins, particularly eIF4AIII, which will be independent of any deposition of the exon junction complex.

The EJC core proteins are well conserved in *Drosophila*; where Y14, MAGO and MLN51 are also known as Tsunagi, Mago Nashi and Barentsz, respectively (*Macchi et al., 2003*; *Mohr et al., 2001*; *Palacios et al., 2004*). The genes encoding these four proteins are all required for oskar mRNA localization to the posterior end of the oocyte (*Macchi et al., 2003*; *Mohr et al., 2001*; *Palacios et al., 2004*; *van Eeden et al., 2001*). Specifically, similar to that reported in mammalian cells, the EJC consisting of eIF4AIII-MAGO-Y14 (MLN51 is mostly cytoplasmic) is deposited at the canonical position 20–24 nt upstream of exon junctions on in vitro spliced mRNAs, and both splicing and deposition of the EJC appear to be required for oskar mRNA localization during oogenesis in *Drosophila* (*Ghosh et al., 2010*, *2012*; *Hachet and Ephrussi, 2004*). Only some introns appear to trigger EJC-dependent nonsense-mediated mRNA decay (NMD) in *Drosophila* (*Gatfield et al., 2003*; *Saulière et al., 2010*). These observations in *Drosophila* and recent reports that the EJC is not present at all exon junctions or solely at canonical positions in mammalian cells, raise the possibility that either deposition or stability of the EJC on spliced mRNA might be a regulated process (*Mühlemann, 2012*; *Saulière et al., 2012*; *Singh et al., 2012*). Additionally, EJC deposition on partially spliced pre-mRNA might modulate splicing of flanking introns in *Drosophila*, yet only for a subset of transcripts (*Ashton-Beaucage et al., 2010*; *Hayashi et al., 2014*; *Malone et al., 2014*; *Roignant and Treisman, 2010*).

With the aim to understand the mechanism that regulates EJC deposition, we used the giant polytene chromosomes from *Drosophila* salivary glands. Similar to other eukaryotes, pre-mRNA splicing occurs co-transcriptionally in *Drosophila* (*Khodor et al., 2011*; *LeMaire and Thummel, 1990*; *Osheim et al., 1985*). Therefore, the polytene chromosomes provide an ideal system to visualize and analyze the mechanism of the association of EJC proteins with both pre-mRNA and nascent spliced transcripts. The data we show here reveal that deposition of the EJC proteins eIF4AIII, Y14 and MAGO on nascent transcripts, neither depends on the presence of introns nor requires the

spliceosomal protein CWC22 in this organism. Additionally, ChIP-seq analysis of Y14 similarly indicates that this protein associates with transcriptionally active genes in *Drosophila* S2 cells independently of splicing.

## Results

### EJC proteins associate with the nascent transcripts

Using antibodies against eIF4AIII, MAGO and Y14, which detect the proteins with minimal cross-reactivity in Western blotting (*Figure 1—figure supplement 1A and B*), we found that EJC proteins are present in both nuclear and cytoplasmic fractions (*Figure 1—figure supplement 1C*). The absolute amounts of these proteins are comparable between the nucleus and cytoplasm, but as indicated by whole salivary gland immunostaining, they are more concentrated in the nucleus (*Figure 1—figure supplement 1D*). On the polytene chromosomes, the signals of all three proteins are most prevalent at transcriptionally active sites, which correspond to distinct cytologically decondensed segments (interbands) of the chromosome (*Figure 1A*). This localization is apparent by simultaneously inspecting the intensity profile of the EJC proteins and DAPI signals along the same chromosome segment (*Figure 1A* rightmost panel); these line profile plots show apparent complementarity of eIF4AIII, Y14 and MAGO signals with DAPI signal (*Figure 1A, III, VI* and *IX*). The signals of the EJC proteins colocalize with those of RNA Pol II (*Figure 1—figure supplement 2A*, the antibody H5, a marker of transcription elongation recognizes the Ser2 hyperphosphorylated CTD of the largest Pol II subunit (*Buratowski, 2009*; *O'Brien et al., 1994*). The association of the EJC proteins with polytene chromosomes is predominantly with nascent transcripts as the signals are RNase sensitive (*Figure 1B*). There is some residual signal of EJC, which cannot be attributed to incomplete RNA digestion, since that of Hrb87F, the hnRNPA1 homologue in *Drosophila* (*Lakhotia et al., 2012*), is completely removed from the chromosomes (*Figure 1B*). The banding pattern of eIF4AIII at polytene chromosomes is different from that of Y14 and MAGO. While the eIF4AIII signal is detected at every Pol II transcription site (*Figure 1—figure supplement 2A*), there are sites at which Y14 and MAGO signals are either absent or just detectable (yellow arrows in *Figure 1—figure supplement 2B, IV-IX*). As all three EJC antibodies used in the present study were raised in rabbits, we used transgenic flies expressing Y14 and eIF4AIII double tagged with HA and FLAG to analyze further the extent of colocalization of the EJC proteins (Material and methods). The tagged proteins are of the predicted size (*Figure 1—figure supplement 3A and B*) and show a chromosomal banding pattern very similar to that of the endogenous proteins (*Figure 1—figure supplement 3C*). Double immunostaining of tagged Y14, and endogenous MAGO shows complete colocalization along the chromosomes (*Figure 1—figure supplement 4*), suggesting that the two proteins are in close association at transcription sites, most likely forming a stable heterodimer, as previously observed in vitro (*Lau et al., 2003*). In agreement with the staining patterns of endogenous Y14 and MAGO differing from that of eIF4AIII, Y14 is either absent from, or just detectable, at many sites at which there is a very strong signal for tagged eIF4AIII (*Figure 1—figure supplement 5*). Cumulatively, our data indicate that the chromosomal binding pattern of Y14 and MAGO differs from that of eIF4AIII, which instead appears to bind at all Pol II transcription sites. In view of the EJC model, at this stage the data could therefore have been interpreted to signify that either Y14 and MAGO associate with eIF4AIII at a later stage, perhaps post-transcriptionally, or as envisaged by previous reports (*Saulière et al., 2010*), that the EJC is not as stable in *Drosophila* as in mammalian cells.

### Association of EJC proteins with nascent transcripts on polytene chromosomes is intron-independent

The presence of eIF4AIII at all transcription sites was particularly surprising, as many of these sites must correspond to intronless genes; ~20% of *Drosophila* genes do not contain introns (*De Renzis et al., 2007*). Initially, we reasoned that perhaps Y14 and MAGO associate with eIF4AIII selectively, possibly on nascent transcripts of intron-containing genes. To test this hypothesis, we examined the distribution of the EJC proteins at heat shock genes on polytene chromosomes, some of which characteristically do not carry introns (*Lis et al., 1981*). Remarkably, following heat shock, clear accumulation of the EJC proteins was observed at heat-shock puffs (classically denominated by their chromosomal map position) of both intron-containing genes (63B, encoding Hsp90; and 93D,

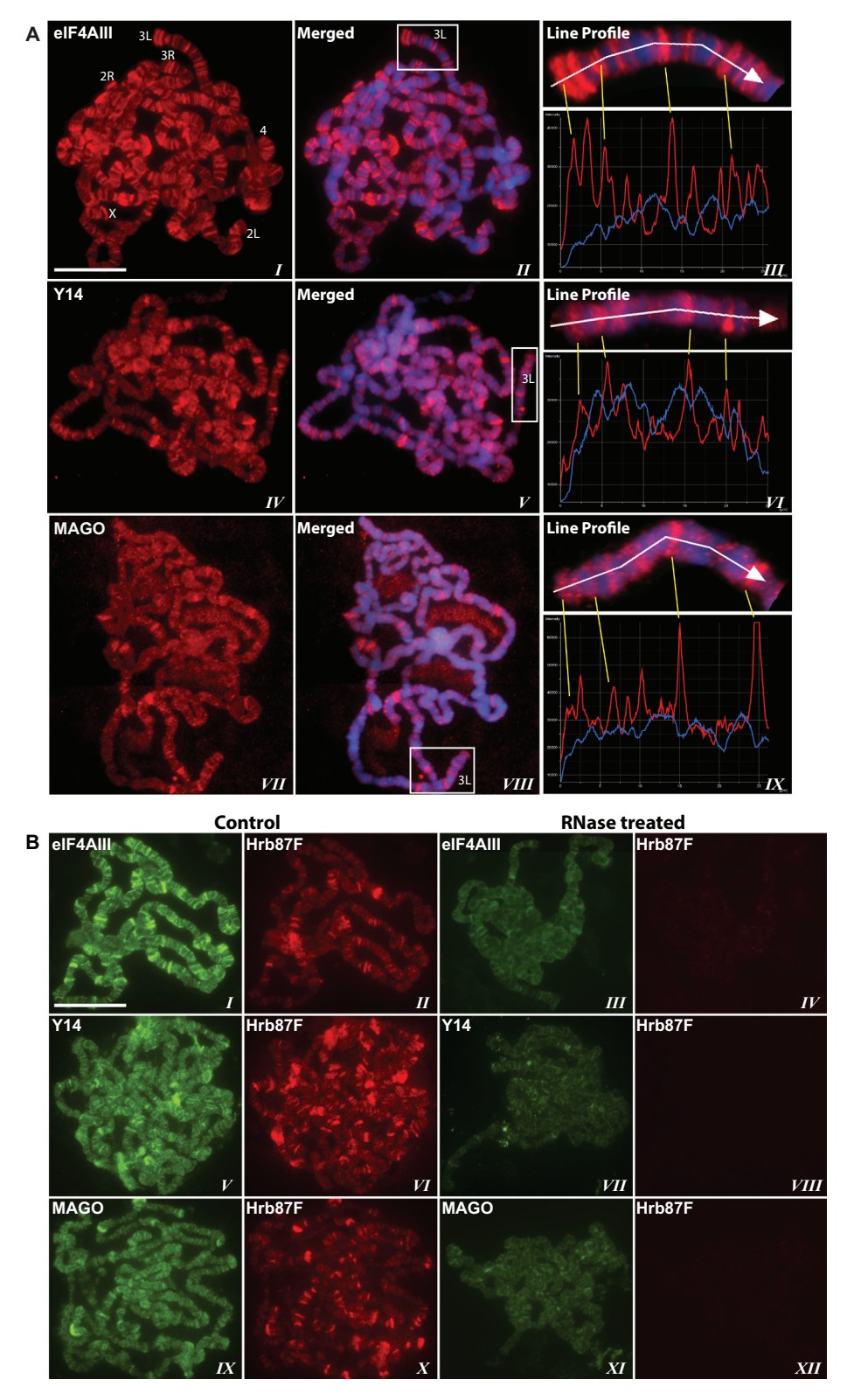

**Figure 1.** The EJC core proteins associate with nascent transcripts at polytene chromosomes. (**A**) Immunolocalization of EJC proteins (red), eIF4AIII (*I-III*), Y14 (*IV-VI*) and MAGO (*VII-IX*), on salivary gland polytene chromosomes of wandering third instar larvae. Chromosomes were counter-stained with DAPI (blue). Intensity profiles of EJC protein and DAPI signals (*III, VI, IX*) over a segment at the tip of chromosome 3L (box in merged images) show accumulation of these proteins at interband regions. Scale bar represents 20 μm length. (**B**) Parallel immunolocalization of EJC proteins (green) and

*Figure 1 continued on next page*

*Figure 1 continued*

Hrb87F (hnRNP A1) (red) on polytene chromosomes spread from salivary gland without treatment (Control) or after incubation with RNase (RNase treated). Scale bar represent 20 μm length.

The following figure supplements are available for figure 1:

**Figure supplement 1.** Characterization of EJC antibodies.

**Figure supplement 2.** EJC protein signals co-localize with active Pol II.

**Figure supplement 3.** Characterization of transgenic flies expressing tagged Y14 or eIF4AIII.

**Figure supplement 4.** Y14 and MAGO strictly colocalize at transcription sites.

**Figure supplement 5.** Y14 banding pattern differs from that of eIF4AIII.

encoding hsrω lncRNAs) as well as that of intronless genes (87A and 87C, encoding Hsp70; and 95D, encoding Hsp68) (*Figure 2*). These observations indicated that interaction of the core EJC proteins with nascent RNA might be splicing independent in *Drosophila*. To assess further whether these proteins associate independently of introns under standard growth conditions at non-heat-shock genes, we constructed a novel inducible expression vector and used it to generate two transgenes, one carrying an intron (S118) and another without intron (S136) (Material and methods). A distinctive feature of these transgenes is that they are flanked by an inducible promoter regulated by an ecdysone responsive element (ERE) (Material and methods). Another feature is that they carry lacO repeats at the upstream of the promoter; these repeats can be easily visualized and cytologically mapped by GFP-lacI immunolocalization on polytene chromosomes (*Robinett et al., 1996*). The transgenes were mapped at position 3B on the X chromosome (S118) and at 63B on 3L chromosome (S136) (*Figure 3A*). The genes are transcriptionally silent in in vitro cultured salivary glands in the absence of ecdysone, but upon ecdysone treatment, a large puff is formed at both loci, indicative of strong transcriptional induction (*Figure 3B*). Notably, immunostaining revealed that eIF4AIII (*Figure 3C*) as well as Y14 (*Figure 3—figure supplement 1*) associate with both transcription puffs, and as expected, there is a strong Pol II Ser2 signal at these loci. These observations therefore indicate that association of the EJC proteins with nascent transcripts can occur independently of splicing in *Drosophila* salivary glands.

## ChIP-seq also indicates intron-independent association of Y14 with active genes in *Drosophila* S2 cells

Next, we used chromatin immunoprecipitation (ChIP) followed by high-throughput DNA sequencing (ChIP-seq) to examine the association of the EJC proteins with gene loci in S2 cells. ChIP experiments were carried out for all three EJC proteins; however, only the Y14 antibody worked well in this assay and produced a clear enrichment profile relative to input DNA; this is most apparent at transcription start sites (TSS) (*Figure 4A*), and selectively with genes that are expressed (*Figure 4B*). Y14 enrichment does not appear to correlate with expression level though. While genes with a very low expression (RPKM between 1 and 10) show lower enrichment, genes ranging from low (RPKM 11–50) to very high expression levels are comparably enriched (*Figure 4—figure supplement 1*). The result of this analysis is therefore consistent with the chromosome immunostaining data shown above, which demonstrated that Y14 and MAGO, unlike eIF4AIII, are either absent or very weak at transcription sites with strong active Pol II signal. Notably Y14 appears to bind both intron-containing and intronless genes (*Figure 4C*), and the enrichment is only slightly higher for genes with introns. When all genes, expressed and unexpressed, are included in the analysis, there is a positive correlation between Y14 binding and intron number (*Figure 4D*); however, in view that Y14 associates selectively with expressed genes, this must signify that many intronless genes are either not or very weakly expressed. A representative genome browser example of this intron-independent association is shown in *Figure 4E*, which corresponds to a region of chromosome three where apparent enrichment is detected at different genes. The highest association is at the *Big brother* (*Bgb*) and

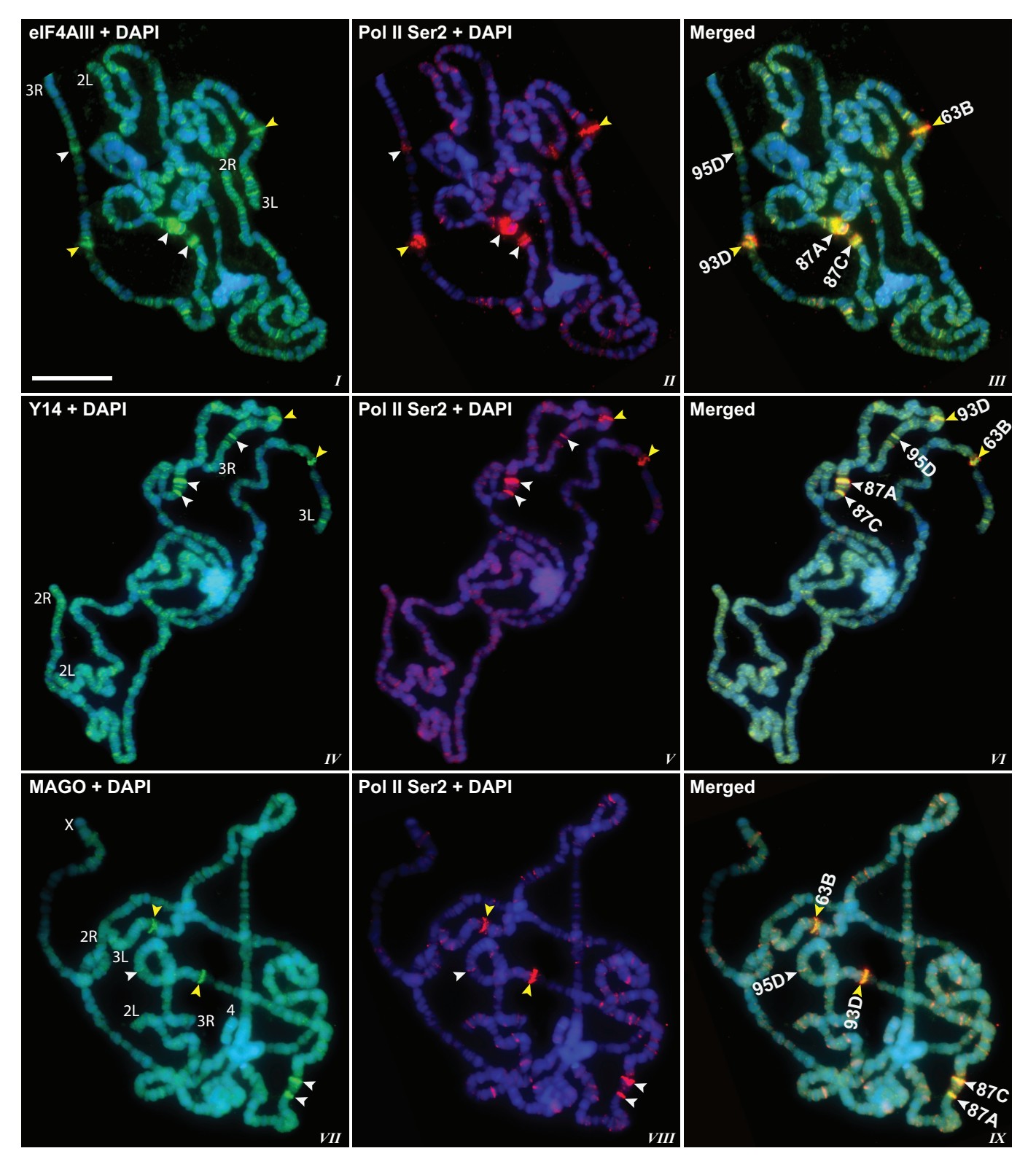

**Figure 2.** EJC proteins accumulate at heat shock transcription puffs. Parallel immunolocalization of hyperphosphorylated RNA Pol II Ser2 (red, *II, III, V, VI, VIII, IX*) and EJC proteins (green) eIF4AIII (*I, III*), Y14 (*IV, VI*), and MAGO (*VII, IX*), on polytene chromosomes following heat shock at 37°C for 1 hr. Yellow arrowheads indicate accumulation of EJC proteins at the induced intron-containing heat-shock genes, while white arrowheads indicate their

*Figure 2 continued on next page*

*Figure 2 continued*

accumulation at intronless heat-shock genes. Genes are identified by their map position (details in Results), which are labeled in the merged images on the right panels. Corresponding chromosome arms are mentioned in left column of panel. Scale bar represents 20 μm length.

*RpL23A* gene loci (indicated by blue and red arrows respectively), both genes are highly expressed in S2 cells, but only *RpL23A* encodes introns. The results of this ChIP-seq analysis are therefore consistent with our observations with polytene chromosomes described above, and further indicate that the association of Y14 with gene loci is dependent on whether they are transcribed, yet unlinked to intron presence. Given that Y14 association with transcription sites is RNA-dependent at the chromosomes (*Figure 1B*), the prediction is that ChIP is detecting mostly Y14 associated with nascent transcript and that the enrichment at TSSs mirrors that of RNA Pol II, which globally in *Drosophila*, as in other metazoan, is detected by ChIP, and other biochemical assays, mostly around TSSs, rather than the gene body (*Adelman and Lis, 2012*; *Core et al., 2012*). However, following up on our observation that association of Y14 (and the other EJC proteins) at the chromosomes is not as sensitive to RNase as hnRNPA1 (*Figure 1B*), we examined whether the EJC proteins might also associate with Pol II directly. We immunoprecipitated (IP) elongating RNA Pol II using a Ser2 CTD antibody (Material and methods). The result was that none of the three EJC proteins make an interaction with Pol II which is stable enough to be detect by IP, instead the association with general elongation factor Spt6 (*Kaplan et al., 2000*), which was used as a positive control, was readily detected (*Figure 4— figure supplement 2*).

## CWC22 is not required for the association of EJC proteins with nascent transcripts

To examine further the connection to splicing, we tested whether the association of the EJC proteins with mRNA depends on CWC22 in *Drosophila*. The prediction from current models is that depletion of CWC22 (known as Nucampholin or NCM in *Drosophila*) would impair association of eIF4AIII with nascent transcripts. RT-PCR quantification shows that CWC22-RNAi efficiently reduces the CWC22 mRNA level (*Figure 5A*). Its depletion in the early stages of development (*fkh-Gal4>NCM-RNAi*) strongly inhibits salivary gland developmental growth and polytenization (*Figure 5B*), to an extent that we were unable to spread the polytene chromosomes for immunolocalization. Due to unavailability of an antibody, the extent of protein depletion could not be assessed directly. The presence of residual CWC22 could not be ruled out. However, the striking growth phenotype indicates that RNAi is efficient at depleting CWC22 essential function. There are no sequences in the *Drosophila* genome which could encode similar proteins, the strong phenotype should therefore indicate that there are also no distant-related proteins that could complement CWC22 function in its absence. Therefore, to asses whether CWC22 might have a role in the association of EJC proteins at the chromosomes, the small-gland phenotype was partially circumvented by inhibiting expression of CWC22-RNAi till early third instar larval stage, by co-expression of the temperature-sensitive Gal80 protein, which represses Gal4 activity when larvae are grown at 18°C (*tub-Gal80^ts*; *Fkh-Gal4>NCM-RNAi*). Larvae were then transferred to 29°C and by the late third instar stage the chromosomes were big enough to proceed for immunostaining. Notably, both eIF4AIII and Y14 remain localized in the nucleus in CWC22 depleted cells (*Figure 5C*, *I-IV* vs. *IX-XII*), and polytene chromosome immunostaining demonstrates that eIF4AIII can still associate with chromosome interbands. Although chromosome morphology is not as well defined in these glands, the banding pattern and relative intensities of the signals are comparable to wild type (*Figure 5C*, *V-VIII*). Additionally, Y14 can also still associate with interbands, producing a staining pattern as intense as, and very similar to, that in wild type (*Figure 5C*, *XIII-XVI*). These observations demonstrate that while CWC22 has an essential function in salivary gland development, it does not appear to be required for the association of eIF4AIII and Y14 (and by inference MAGO) with the nascent transcripts in *Drosophila*.

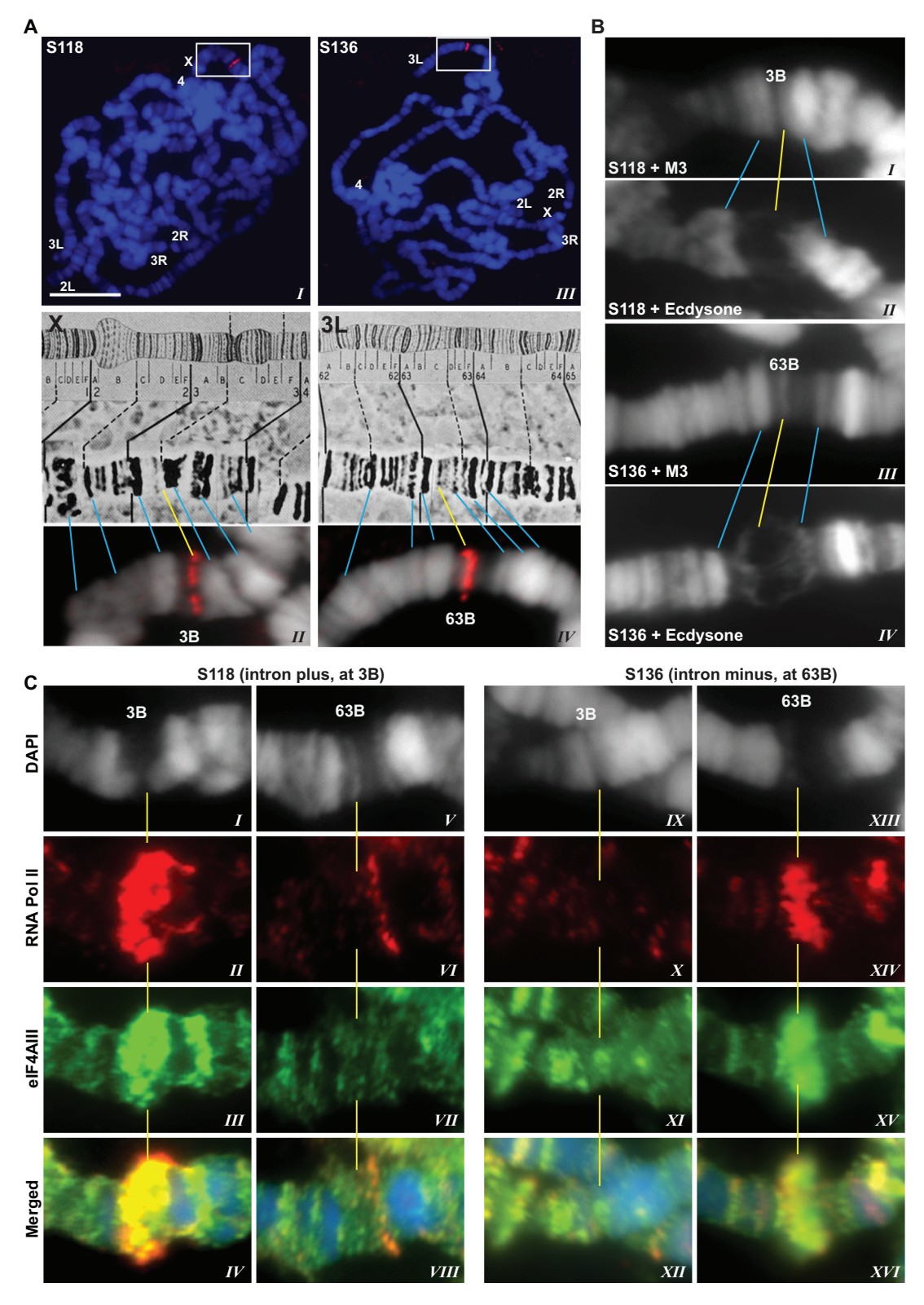

**Figure 3.** EJC proteins associate with nascent transcripts of both intron and intronless genes. (**A**) Immunolocalization of GFP-LacI (red band in boxed areas) at transgene insertion site: S118 (intron plus) on the X (panel *I*) and of S136 (intron minus) on the 3L (*III*) chromosome arm. Bands were mapped at 3B for S118 (*II*) and 63B for S136 (*IV*) using a standard polytene chromosome map shown above. (**B**) DAPI-stained (gray) segments of the X chromosome encompassing 3B (*I, II*) and 3L chromosome encompassing 63B (*III, IV*), without (*I, III*) or with (*II, IV*) ecdysone treatment, which produces a distinct puff at

*Figure 3 continued on next page*

*Figure 3 continued*

the transgene insertion locus. (**C**) Immunolocalization of RNA Pol II Ser2 (red) and eIF4AIII (green) at 3B (*I-IV* and *IX-XII*) and 63B (*V-VIII* and *XIII-XVI*) loci in S118 (*I-VIII*) and S136 (*IX-XVI*) transgene following ecdysone treatment. As there is no insert at locus 63B in S118 (*V-VIII*) and at 3B in S136 (*IX-XII*), these are used as ecdysone-unresponsive control loci for the transgene at 63B in S136 (*XIII-XVI*) and for 3B in S118 (*I-IV*), respectively. Scale bar represents 20 μm length.

The following figure supplement is available for figure 3:

**Figure supplement 1.** Y14 associates with nascent transcript of both intron and intronless gene reporters.

## Association of Y14/MAGO with nascent transcripts does not require eIF4AIII

Since the interaction of Y14/MAGO with eIF4AIII drives RNA binding in mammalian cells and stabilizes the EJC in vitro (*Ballut et al., 2005*), a prediction is that Y14/MAGO may not associate, or does so to a lesser extent, with nascent transcripts in the absence of eIF4AIII. We tested this by visualizing the distribution of Y14 on the polytene chromosomes from salivary glands depleted of eIF4AIII by RNAi (*Figure 6A,B and C*). As for CWC22, depletion of eIF4AIII from an early stage of development (*Fkh-Gal4>eIF4AIII-RNAi*) strongly inhibits salivary gland growth (*Figure 5B*); therefore, indicating that RNAi is effectively depleting an essential function of the protein. Again, the phenotype prevented polytenization of the chromosomes. However, by restricting RNAi activation until early third instar stage, (using *tub-Gal80^{ts}; Fkh-Gal4>eIF4AIII-RNAi*) glands were big enough to prepare satisfactory chromosome spreads for immunostaining (*Figure 6A*). Remarkably, we observed that despite the drastic reduction in chromosomal eIF4AIII signal (*Figure 6A*, *VII* and *VIII* vs. *V* and *VI*), and despite the low level of polytenization, Y14 could still be detected at cytologically distinguishable interbands, similar to wild type (*Figure 6A*, *XV* and *XVI* vs. *XIII* and *XIV*). In contrast to depletion of CWC22 and eIF4AIII which strongly inhibited salivary glands growth, depletion of either Y14 or MAGO resulted in visually normal salivary glands (*Figure 5B*), despite their depletion being very efficient, based on salivary gland Western blotting (*Figure 6D and E*). Although some Y14 signal persists in the nucleolus (which often remains attached to the chromosomes during the spreading procedure), the respective chromosomal signals are completely absent from the chromosome arms (6F, *I* and *II*). Particularly, depletion of neither Y14 nor MAGO affects the association of eIF4AIII with the chromosomes (*Figure 6F, V* and *XI*); we observed instead that depletion of Y14 eliminates MAGO from the chromosomes (*Figure 6F, III* and *IV*) and depletion of MAGO also reduces the Y14 signal (*Figure 6F, IX* and *X*). In summary, these observations indicate that association of Y14 (and by inference MAGO) with nascent transcripts does not require eIF4AIII and that of eIF4AIII is independent of Y14 or MAGO. Additionally, the RNAi phenotype of eIF4AIII is drastically different: while eIF4AIII knockdown impairs salivary gland development, neither that of Y14 nor MAGO has any apparent visual phenotype (*Figure 5B*).

## eIF4AIII persistently associates with actively translating transcripts via interactions with both mRNA and ribosomal subunits

The current understanding is that the EJC remains associated with spliced mRNAs until it is removed by the ribosome in the first round of translation. A key observation supporting this model is that Y14 is present in monosomal but not polysomal fractions in mammalian cells (*Diem et al., 2007*; *Dostie and Dreyfuss, 2002*; *Gehring et al., 2009*). We therefore analyzed the distribution of the EJC proteins in polysomal fractions of *Drosophila* S2 cells. Surprisingly, we detected neither Y14 nor MAGO in monosomal fractions; both proteins remain on the top of the gradient, corresponding to free proteins or small molecular weight complexes (*Figure 7A*). In contrast, eIF4AIII was detected in all the fractions, but it was least abundant in the fraction containing Y14 and MAGO (*Figure 7A*). There is a small correlation between eIF4AIII levels and the ribosomal trace; however, eIF4AIII is no more abundant in either monosomal or lighter fractions, it is therefore unlikely that more of it is associated with mRNAs at or before the first round of translation when they are loaded with just one ribosome or with the 40S ribosomal subunit. Such distribution suggests that eIF4AIII is not removed by translation and remains associated actively translated mRNA; this interpretation is further

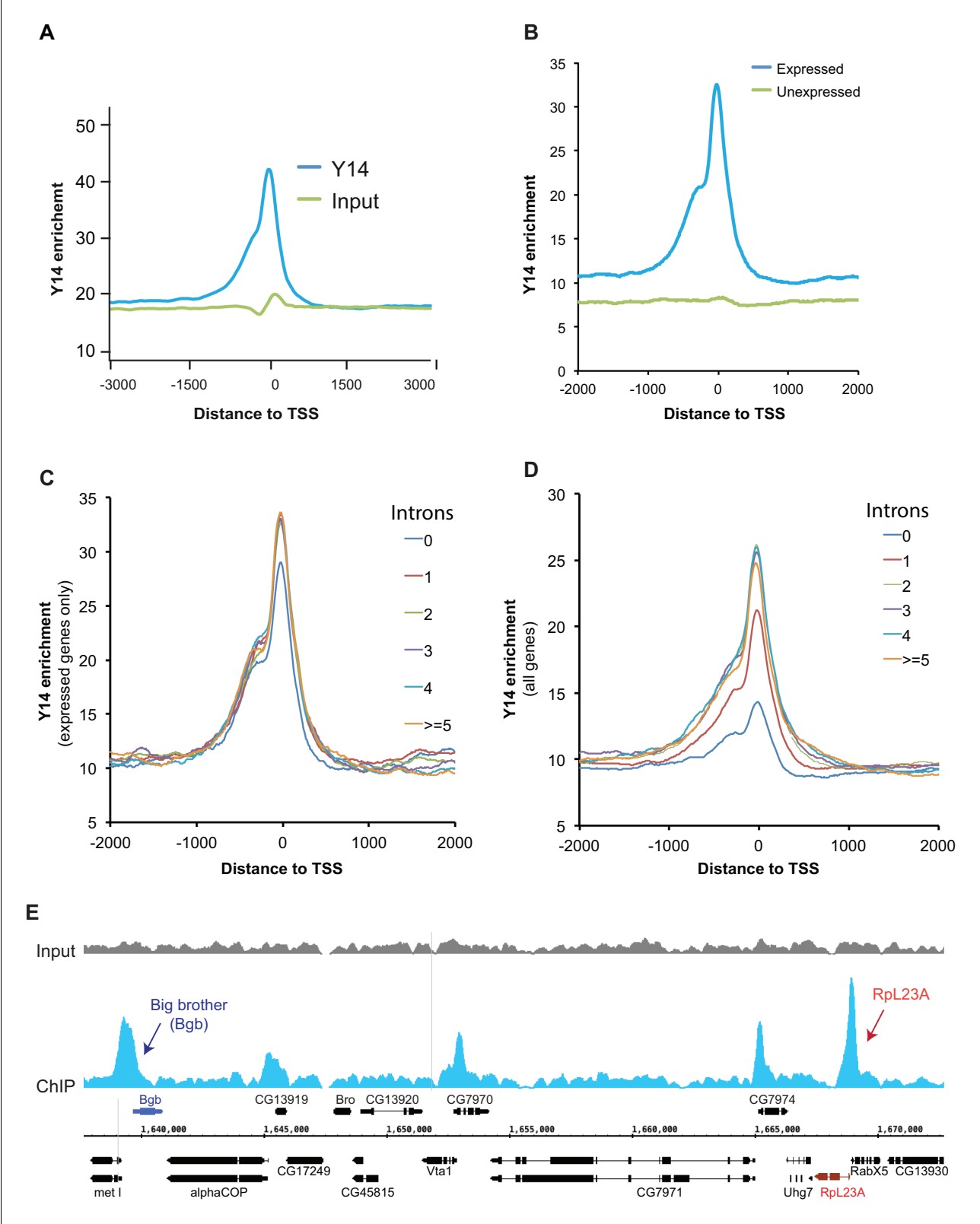

**Figure 4.** Y14 associates with expressed genes independent of introns in S2 cells. (**A**) Genome-wide average Y14 ChIP-seq enrichment 3000 bases around transcription start sites (TSS) of ChIP (blue trace) versus input DNA (green). (**B**) Y14 ChIP-seq enrichment (after background subtraction) 2000 bases around transcription start sites (TSS) of expressed (blue trace) or unexpressed (green trace) genes in S2 cells (see Material and methods). (**C**) Average Y14 enrichment of expressed genes with a different number of introns: 0 to >=5, indicated by traces of different colour (see legend on right of

*Figure 4 continued on next page*

*Figure 4 continued*

the plot). (D) Average Y14 enrichment across genes as in C but including both expressed and unexpressed genes. (E) Representative chromosome region showing Y14 ChIP-seq enrichment profile (blue), versus that of input DNA (grey). Genes are labeled following Flybase nomenclature.

The following figure supplements are available for figure 4:

**Figure supplement 1.** Y14 association with transcribed genes does not correlate with mRNA levels.

**Figure supplement 2.** EJC proteins do not co-purify with RNA Pol II.

substantiated by the fact that mild RNase treatment, which breaks down polysomes by digesting the mRNA (Material and methods), shifts most of eIF4AIII into lighter sub-ribosomal fractions (*Figure 7B*). The sedimentation of Y14 and MAGO is not affected by the RNase treatment. In vitro dissociation of polysomes by EDTA treatment also shifts most of eIF4AIII in the lightest fraction (*Figure 7C*, lane 12), indicative of both polysome breakdown and perhaps release of the protein from mRNA. Notably, a portion of eIF4AIII apparently co-fractionates with the 60S ribosomal subunit (*Figure 7C*, lanes 9 and 10), and seemingly to a lesser extent with the 40S (fraction 11 in *Figure 7C*; the red arrow indicates a faster migrating form of the protein in lane 12, which possibly was generated by proteolysis during the in-vitro incubation). Some of eIF4AIII might fractionate with ribosomal subunits also in the untreated sample (*Figure 7A*, fractions 9 and 10). We investigated further whether there is an association with ribosomal subunits by pre-treatment of the cell culture with puromycin, followed by dissociation of the subunits by incubation at either 4°C or room temperature in presence of high-salt in vitro prior to loading on the gradient (Material and methods). This treatment also delocalized eIF4AIII to lighter fractions, consistent with most of the protein being associated with actively translated mRNA. The majority of the protein was detected in the top two light fractions (*Figure 7D*, lanes 11 and 12; as seen in *Figure 7C*, a large fraction of the protein runs as a faster migrating photolytic cleavage product). Cumulatively, the data demonstrate that eIF4AIII cosediments with both monosomes and polysomes, yet Y14 and MAGO fractionate separately in normal conditions, and therefore, it is not likely they form a stable complex with eIF4AIII on translating mRNA in S2 cells. Although some eIF4AIII is detected in the light fraction containing Y14 and MAGO, this is a relatively very minor amount of what is found in the heavier fractions (possibly the result of some unavoidable mixing during fractionation).

## Y14 and MAGO are required for the stability of each other

As the signals for Y14 and MAGO completely overlap at the chromosomes (*Figure 1—figure supplement 4*), and depletion of either protein affects the association of the other (*Figure 6F*), we predicted that dimerization is most likely stabilizing these two proteins. To further investigate this, we assayed protein levels following RNAi-mediated knockdown of Y14, MAGO and eIF4AIII in *Drosophila* S2 cells. Notably, consistent with what we observed at the chromosomes, knockdown of Y14 drastically reduces MAGO, and conversely, MAGO knockdown reduces the level of Y14, although not completely (*Figure 8A*). In both cases, the level of eIF4AIII was not affected. Furthermore, while knockdown of eIF4AIII led to an overall reduction in protein levels, it did not noticeably affect Y14 or MAGO. The inter-dependence is most likely the result of a change in stability of the proteins, as depletion of Y14 does not affect MAGO mRNA level and that of MAGO does not affect Y14 mRNA level (*Figure 8B*). These results indicate that Y14 and MAGO heterodimerization is essential for their stability in cells– consistent with what was reported by one of the initial EJC studies (*Le Hir et al., 2001*)- yet it appears such effect is independent of eIF4AIII.

## Discussion

Our investigation reveals that the three proteins that are thought to form the core of the EJC across organisms, associate with nascent transcripts independently of intron presence and of CWC22 in *Drosophila*, the spliceosome component predicted to drive the assembly of the EJC on spliced mRNAs in mammalian cells (*Alexandrov et al., 2012*; *Barbosa et al., 2012*; *Saulière et al., 2012*; *Singh et al., 2012*; *Steckelberg et al., 2012*). Our data indicate that Y14 and MAGO form an

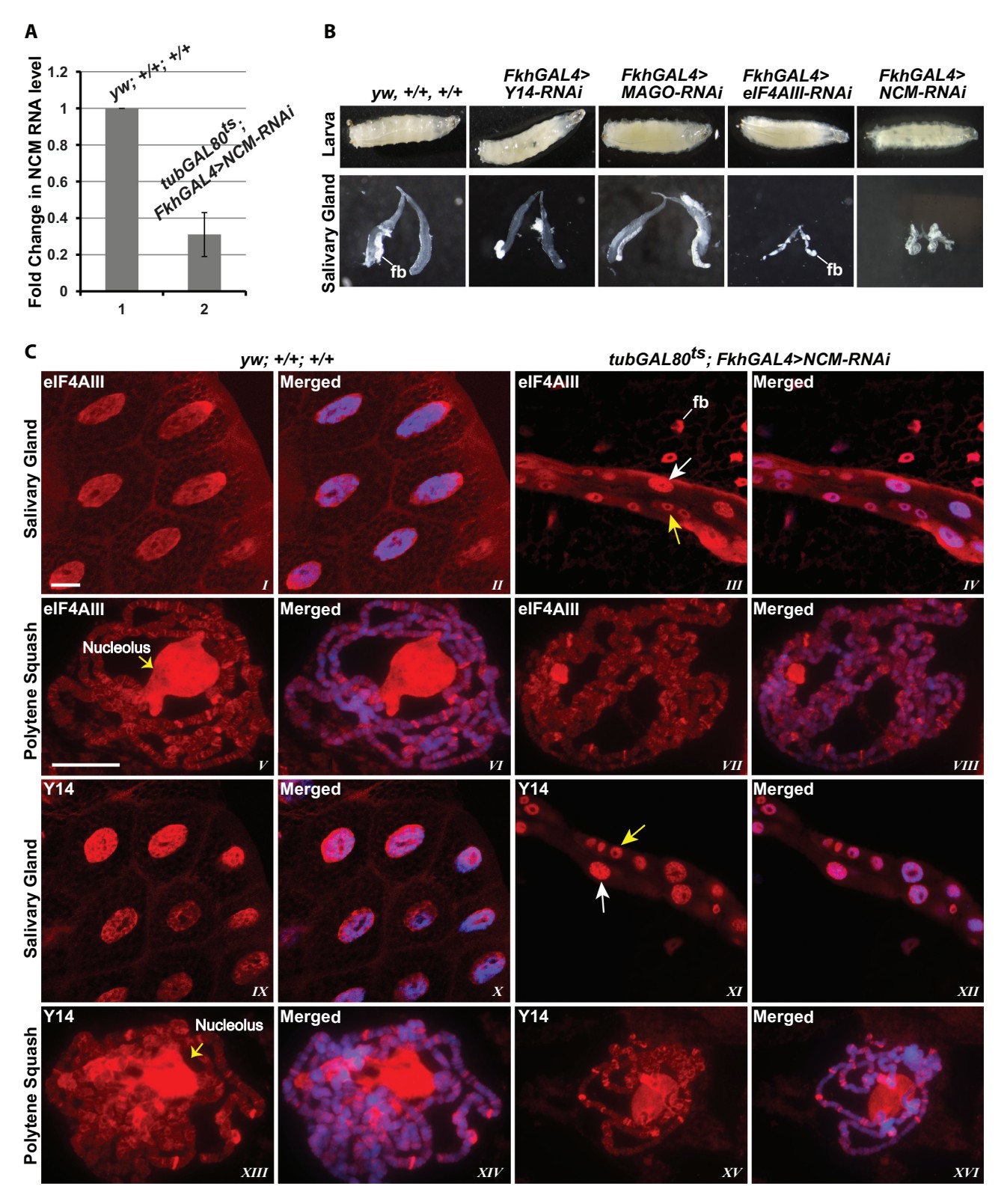

**Figure 5.** EJC proteins associate with nascent transcripts independently of CWC22 (NCM). (**A**) Real-time PCR quantification of NCM RNA level in salivary glands of *tubGAL80^ts; +; FkhGAL4>NCM-RNAi* (right) relative to that in wild type (left). (**B**) Thirst instar larva and their salivary glands of different genotypes mentioned above each lane. The line indicates a fragment of fat body (fb) adhering to the glands. (**C**) Immunolocalization of EJC proteins eIF4AIII and Y14 in whole salivary gland cells (*I-IV, IX-XII*) and at polytene chromosomes (*V-VIII, XIII-XVI*). The two panels on the left are from wild-type

*Figure 5 continued on next page*

Figure 5 continued

larvae and the two on the right are from *tubGAL80^{ts}; +; fkhGAL4/NCM-RNAi* larvae. In the mutant (*III, IV, XI, XII*) white arrows indicate relatively large-sized nuclei while yellow arrows indicate small-sized nuclei. The white line (panel *III*) indicates a fat body (fb) cell nucleus attached with salivary gland. Chromosomes were counterstained with DAPI (blue). Scale bars represent 20 μm length for corresponding set of images.

obligatory heterodimer in vivo, as observed previously (*Lau et al., 2003*; *Le Hir et al., 2001*). Y14/MAGO associates with nascent transcripts on polytene chromosomes, yet this association neither correlates with, nor requires eIF4AIII, suggesting that the Y14/MAGO heterodimer engages with nascent mRNAs independently of eIF4AIII. Moreover, Y14 and MAGO do not cosediment with eIF4AIII by sucrose gradient fractionation of S2 cell cytoplasmic extracts, indicating that these proteins are not forming a stable complex with eIF4AIII. As Y14 and MAGO stay on the top of the gradient, it is most likely that they are not bound to mRNA in the cytoplasm. In contrast, eIF4AIII associates with mRNA loaded with multiple ribosomes; it does not therefore appear to be removed, at least not irreversibly, by translocating ribosomes. Perhaps eIF4AIII primarily binds mRNA UTRs in *Drosophila* cytoplasmic extracts, and, to a smaller extent, as indicated by our sucrose fractionation data, ribosomal subunits, rather than coding regions, as reported for mammalian whole-cell lysates (*Saulière et al., 2012*; *Singh et al., 2012*). Particularly, our data indicate that Y14-MAGO dissociates from mRNA before translation initiation, unlike in mammalian cells (*Diem et al., 2007*; *Dostie and Dreyfuss, 2002*; *Gehring et al., 2009*). This interpretation is consistent with the report that Partner of Y14-MAGO (PYM) does not engage with the ribosome and that PYM-mediated dissociation of EJC proteins from oskar mRNA is independent of translation in ovarian extracts in *Drosophila melanogaster* (*Ghosh et al., 2014*).

We speculate that in *Drosophila* eIF4AIII, similar to related RNA helicases, dynamically binds mRNA in an ATP-driven reaction rather than making a stable interaction with a specific location on the RNA (*Linder and Jankowsky, 2011*). Our data suggest that eIF4AIII is a general mRNP component, and it may therefore have a global role in pre-mRNA processing and translation, independently of splicing and EJC assembly in *Drosophila*. The protein shares extensive sequence and domain similarity with the essential eukaryotic translation initiation factor eIF4A across eukaryotes (*Parsyan et al., 2011*), which is its closest homolog in *Drosophila* with 71% aminoacid sequence identity. The protein eIF4A is the prototype of an RNA helicase which is required for unwinding 5'UTR secondary structures during the initial recruitment of the ribosome to mRNA 5' end (*Linder and Jankowsky, 2011*). Consistent with this role, eIF4A was reported to sediment in lighter sucrose fraction corresponding to mRNAs not yet engaged in translation elongation (*Bordeleau et al., 2006*). However, as we find eIF4AIII in heavy polysomal fractions, we predict that the protein might mostly bind 3'UTRs rather than 5'UTRs of polysomal mRNA. What the significance is of the partial cosedimentation of eIF4AIII with ribosomal subunits will need to be investigated. In this respect, eIF4AIII might be similar to VASA, another eIF4A-related protein that is required for germline differentiation in *Drosophila*; VASA appears to regulate translation via its association with specific 3'UTRs (*Liu et al., 2009*). While it has long been understood that human eIF4AIII is functionally distinct from eIF4A (*Li et al., 1999*; *Parsyan et al., 2011*), it was recently reported that the protein enhances translation of mRNAs associated with the nuclear cap binding complex (CBC) in human cells, independently of the presence of introns (*Choe et al., 2014*). As depletion of eIF4AIII impairs salivary gland growth, we predict that its essential role in *Drosophila* development might primarily reflect a global requirement in translation, rather than localization of specific mRNAs (*Palacios et al., 2004*).

Depletion of the EJC proteins causes changes in splicing patterns in *Drosophila* and other organisms (*Ashton-Beaucage et al., 2010*; *Hayashi et al., 2014*; *Malone et al., 2014*; *Michelle et al., 2012*; *Roignant and Treisman, 2010*; *Wang et al., 2014*). These changes can be interpreted in two ways: either canonical EJC deposition can influence splicing of neighbouring introns or the EJC proteins can affect splice site recognition by being functional components of pre-spliceosome intermediates. Both interpretations are based on the assumption that EJC assembly is either constitutively coupled to splicing in *Drosophila*, as reported (*Ghosh et al., 2012*), or that it is at least deposited on a subset of spliced mRNAs or pre-mRNAs. Our data are inconsistent with these interpretations. We cannot exclude the possibility that the EJCs may be deposited at specific junctions

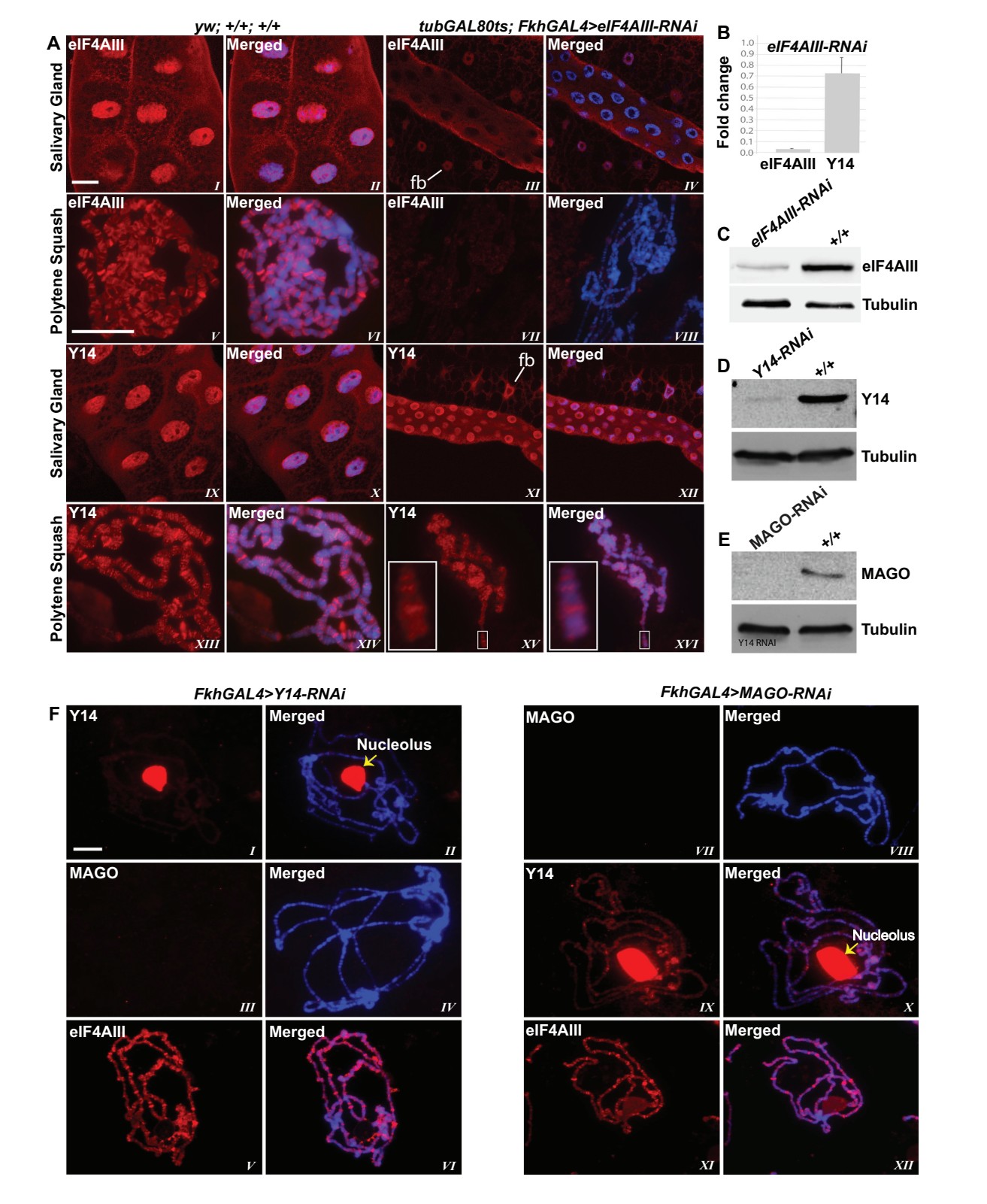

**Figure 6.** eIF4AIII is not required for the association of Y14 and MAGO with nascent transcript. (**A**) Immunolocalization of EJC proteins (red) eIF4AIII and Y14 in whole salivary gland (*I-IV, IX-XII*) and polytene chromosomes (*V-VIII, XIII-XVI*), in wild type (left panels) and *tubGAL80ts; +; fkhGAL4/eIF4AIII-RNAi* (right panels) larvae. Insets (*XV, XVI*) are showing a magnified view of the areas enclosed by the white boxes. Chromosomes were counterstained with DAPI (blue). Scale bar represents 20 μm length. (**B**) Real-time PCR quantification of eIF4AIII and Y14 mRNA levels in *fkhGAL4>eIF4AIIIRNAi* relative

*Figure 6 continued on next page*

Choudhury *et al.* eLife 2016;5:e19881. DOI: 10.7554/eLife.19881                                                         14 of 24

*Figure 6 continued*

to wild-type larval salivary glands. (C) Western blotting showing eIF4AIII protein levels in *tubGAL80^{ts}; +; fkhGAL4/eIF4AIII-RNAi* and wild-type glands. (D) Western blotting showing Y14 protein level in *fkhGAL4>Y14RNAi* and wild-type salivary glands. (E) Western blotting showing level of MAGO protein in *fkhGAL4>MAGO-RNAi* and wild-type salivary glands. Tubulin was detected as loading control. (F) Immunolocalization of Y14 (red, *I, II, IX, X*), MAGO (red, *III, IV, VII, VIII*) and eIF4AIII (red, *V, VI, XI, XII*) on polytene chromosomes from *fkhGAL4>Y14RNAi* (left panels) and *fkhGAL4>MAGO-RNAi* (right panels) larvae. Chromosomes were counterstained with DAPI (blue). Yellow arrows (*II, X*) indicate accumulation of corresponding proteins at the nucleolus. Scale bar represents 20 μm length.

The following figure supplement is available for figure 6:

**Figure supplement 1.** Depletion of eIF4AIII does not affect the pattern of recruitment of Y14 at Pol II transcription sites.

on some mRNAs, and that these are stable enough to persist through nuclear export and cytoplasmic mRNA localization in some cells, oocytes in particular (*Ghosh et al., 2010, 2012; Hachet and Ephrussi, 2004*). While the Y14 ChIP-assay shows slightly higher association with genes containing multiple introns (*Figure 4D*), this might be a consequence of their higher transcription rate, rather than being driven by a direct interaction with spliceosome components. The most parsimonious explanation is that the so-called EJC proteins bind nascent transcripts independently of splicing and eIF4AIII, and therefore may not form a stable complex which can tag splice junctions, or even spliced mRNPs, on *Drosophila* mRNAs. Therefore, the mechanisms by which these proteins regulate pre-mRNA splicing, transposon activity, mRNA localization and translation possibly need to be re-examined in the context of the absence of an EJC.

A well-characterized function of the EJC is its role in linking pre-mRNA splicing to translation and NMD in mammalian cells (*Chazal et al., 2013; Le Hir and Séraphin, 2008; Saltzman et al., 2008*). While this might apply to some *Drosophila* introns (*Saulière et al., 2010*), it has instead been reported that splicing does not affect NMD of well-characterized mRNA reporters in *Drosophila* (*Gatfield et al., 2003*). Additionally, deposition of the EJC, or a similar complex which would tag splice junctions, does not appear to be the mechanism that links pre-mRNA splicing to NMD in fission yeast (*Wen and Brogna, 2010*). Therefore, the important question of how splicing can affect translation and NMD remains to be understood (*Brogna et al., 2016; Brogna and Wen, 2009*).

## Material and methods

### *Drosophila* strains

Flies were reared in standard fly food medium at 24°C. The yw strain was used as wild type (DGGR_108736). Double tagged (2X HA-FLAG) Y14 and eIF4AIII transgenes were generated by cloning the corresponding cDNA sequence into pUAST-attB vector and insertion at the PhiC31 recombination site of the yw, P{CaryPattP3} strain using germline injection (BestGene, USA). Transgenes expressing the lacO-tagged and ecdysone-regulated S118 and S136 constructs (see below) were generated by random P-element transformation. RNAi lines targeting Y14 (36585) and eIF4AIII (32444) were obtained from the Bloomington stock centre while that of MAGO (28132) was obtained from the VDRC stock centre. The GFP-LacI stock was previously described (*Vazquez et al., 2002*). The *forkhead* (*fkh*)-*Gal4* used in present study has a salivary-gland-specific expression from early stage of development (*Henderson and Andrew, 2000*). The *tubGal80^{ts}* line was previously described (*McGuire et al., 2003*). Flies with either the *tubGAL80^{ts}/+; +; fkhGAL4/eIF4AIII-RNAi* or *tubGAL80^{ts}/+; +; fkhGAL4/NCM-RNAi* genotype were obtained by selecting tubby female larvae from crossing *tubGAL80^{ts}; +; fkhGAL4/TM6B* male with either *+; eIF4AIII-RNAi* or *+; NCM-RNAi* female flies, respectively. These larvae were initially cultured at 18°C until early third instar stage and then transferred at 29°C until late third instar larval stage when salivary glands were dissected out.

### Antibodies

Affinity purified antibodies against *Drosophila* EJC proteins, anti-MAGO, anti-Y14 and anti-eIF4AIII were produced in rabbit using recombinant proteins expressed in *E. coli* by YenZym (USA). Their dilution used in immunostaining was typically 1:100 while it was 1:1000 in Western blotting. Other

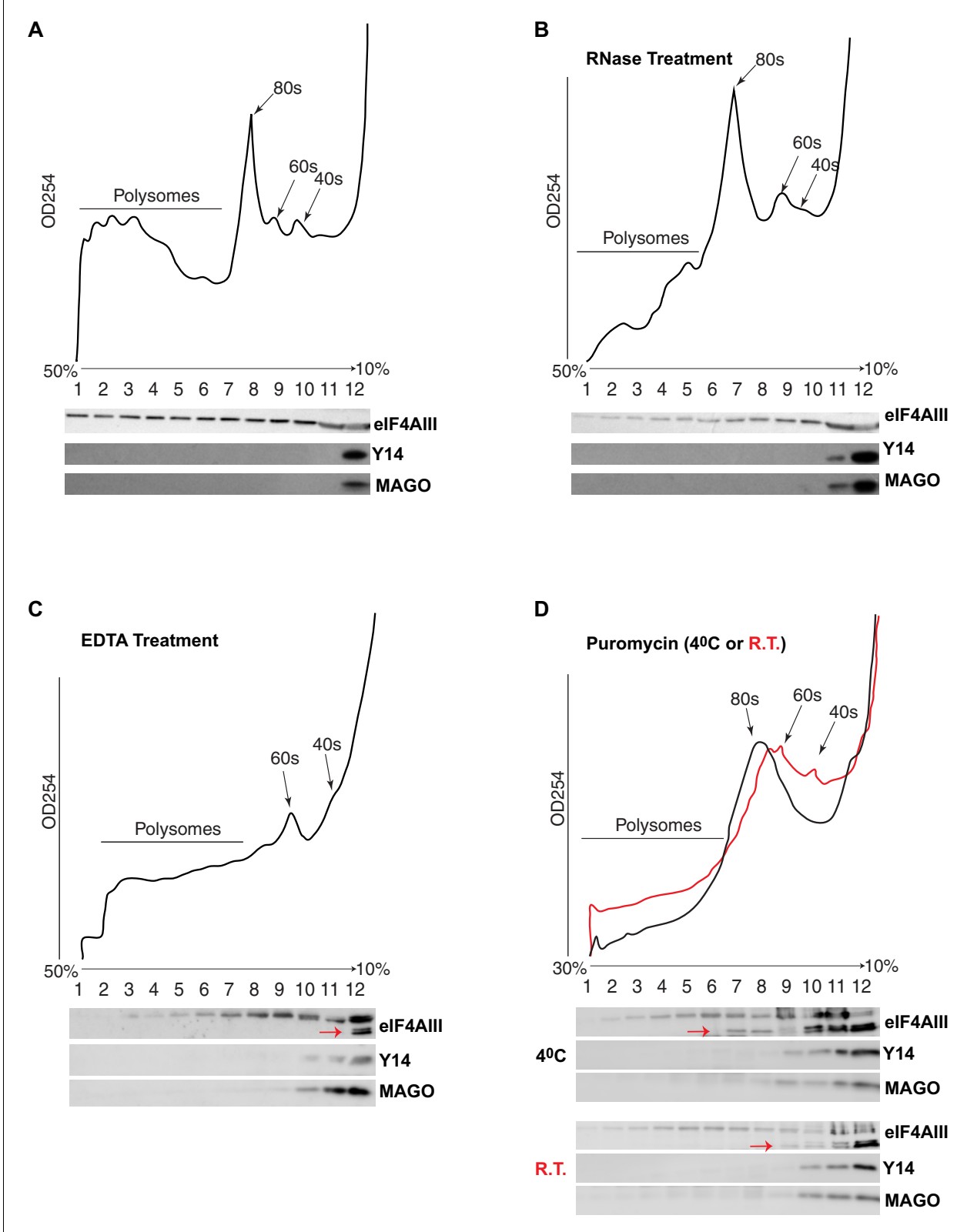

**Figure 7.** eIF4AIII but not Y14-MAGO associates with ribosome-loaded mRNA in S2 cells. (**A**) Polysome profiling of cytoplasmic extracts of S2 cells, and Western blotting of corresponding fractions, show distribution of eIF4AIII (upper lane), Y14 (middle lane) and MAGO (lower lane). (**B**) Polysome profiling and Western blotting as shown in A, following RNase treatment. (**C**) Polysome profiling and Western blotting following EDTA treatment. (**D**)
*Figure 7 continued on next page*

*Figure 7 continued*

Polysome profiling and Western blotting following puromycin treatment at either 4°C (black line) or at room temperature (red line). The red arrows in C and D point to a faster migrating double band of eIF4AIII.

primary antibodies used in present study, and their dilution in immunostaining were mouse anti-Pol II (AB_10143905, H5, Covance, 1:250), mouse anti-HA (AB_514505, 12CA5, 1:500), mouse anti-FLAG (AB_259529, M2, Sigma-Aldrich, 1: 200) and goat anti-GFP (Bio-Rad AbD Serotec, 1:250). The antibodies used in Western blotting were diluted as follow: anti-RNA Pol II (AB_10013665, 8WG16, Covance, 1:2000), anti-Pol II Ser2 (3E10, Merck Millipore, 1:1000), Anti-Spt6 (1:1000), mouse anti-HA (AB_514505, 12CA5, 1:5000), mouse anti-FLAG (AB_259529, M2, Sigma-Aldrich, 1: 2000), mouse

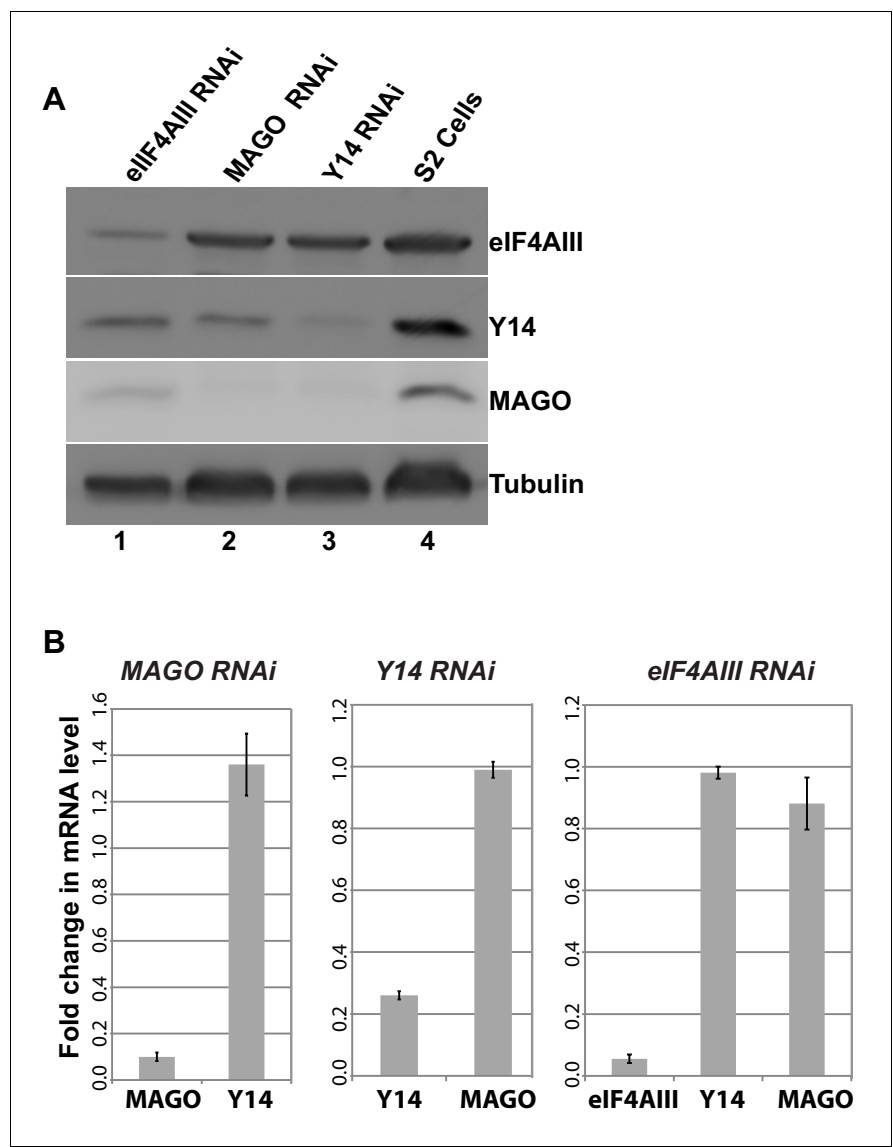

**Figure 8.** Y14 and MAGO are required for the stability of each other. (**A**) Western blotting of total cell lysate from S2 cells to detect eIF4AIII, Y14, MAGO, following eIF4AIII-RNAi, MAGO-RNAi, Y14-RNAi and in untreated cultures. Tubulin was detected as loading control. (**B**) Real-time PCR quantification of the mRNA level of indicated transcripts was carried out in MAGO-RNAi (left), Y14-RNAi (middle) and eIF4AIII-RNAi (right) along with untreated S2 cells.

anti-Hrb87F (hnRNPA1) (1:200) and mouse anti-β-tubulin (Sigma-Aldrich, T5168 1:4000). Secondary antibodies were from Jackson Immuno Research Technologies, Sigma-Aldrich or Life Technologies.

## Ecdysone-inducible lacO-tagged constructs

The intron-containing construct (S118) was generated by cloning medfly *adh1* coding region including the intron (*Brogna et al., 2006*) in the EcoRI site of the pERE expression and germline transformation vector described below. This construct carries six repeats of the MS2 binding site in the 3′ UTR, subcloned from pIG-bs6-mix (*Golding and Cox, 2004*). The intronless construct (S136) encodes a fusion transcript coding for 3 HA repeats, enhanced green fluorescent protein (EGFP) and beta-galactosidase (LacZ). In pERE ecdysone-dependent transcription is driven by a promoter cassette consisting of approximately seven repeats of the ecdysone responsive element (ERE) from *hsp27* (*Riddihough and Pelham, 1987*); this was cloned upstream of the *Adh* distal promoter, both from *Drosophila* (a clone of this cassette which was provided by Carl Thummel, University of Utah, has been generated and previously described (Steve Stowers, PhD thesis, Stanford University). To generate pERE, the cassette was cloned into the HindIII and EcoRI sites of pUAST, therefore replacing the UAS promoter with the ERE. Additionally, eight lacO repeats [*Robinett et al., 1996*]) were cloned into the HindIII and PstI sites located just upstream of the ERE cassette to allow visualization of the locus using GFP-LacI.

## Heat shock and ecdysone induction

For heat-shock treatment, actively wandering third instar larvae of desired genotypes were transferred in batches to microfuge tubes lined with moist filter paper and submerged in a water bath maintained at 37°C for 45 min. Control samples of larvae of comparable age and genotypes were kept in microfuge tubes containing moist filter papers at 24°C, in parallel. For ecdysone treatment, salivary glands of actively wandering third instar larvae were dissected out and incubated in Shields and Sang M3 insect media (M3, Sigma) with or without 1 µM ecdysone, for 1 hr at room temperature.

## Cell culture and RNA interference

*Drosophila* embryo driven S2 cells (CVCL_Z232) were cultured in Insect–XPRESS medium (Lonza) supplemented with 10% fetal bovine serum (FBS) and 1% Penicillin-Streptomycin-Glutamine mix (P/S/G) (Invitrogen) at 27°C. Y14 or eIF4AIII sequence was PCR amplified with corresponding primer pairs from available plasmid constructs, while that of MAGO was amplified from adult male flies (*Supplementary file 1*). Along with the desired gene sequence, all these primer pairs carried the T7 promoter sequence at their 5′ end (5′-TTAATACGACTCACTATAGGGGAGA-3′). The amplified PCR fragments were purified with a QIAquick PCR Purification Kit (QIAGEN) and dsRNA was synthesized using the T7 RiboMAX express RNAi system (Promega). Typically, a six-well culture dish was seeded with $10^6$ cells/well in serum-free media followed by addition of 15 µg of dsRNA into each well and by 1 hr incubation at room temperature. After the incubation, 2 mL of complete media was added to each well and the cells were incubated for four days to knockdown the corresponding RNA.

## RNA extraction and real-time PCR

RNA extraction from S2 cells was carried out using the RNeasy Mini Kit (QIAGEN). Total RNA (700 ng) was used for cDNA synthesis using qScript cDNA synthesis Kit (Quanta Biosciences). Quantitative real-time PCR was carried out using the SensiFAST SYBR Hi ROX Kit (Bioline) in 96-well plates on a ABI PRISM 7000 (Applied Biosystems). RNA isolation from salivary gland and real time quantification of desired gene transcripts was done with a Power SYBR Green Cells-to-Ct Kit (Thermo Fisher Scientific). The Ct value for the desired transcript level was normalized by the RpL32 or 18 s rRNA transcripts as reference (*Supplementary file 1*).

## Isolation of nuclear and cytoplasmic fractions from S2 cells

Nuclear and cytoplasmic fractions were purified following a published procedure (*Parker and Topol, 1984*). S2 cells were grown in a T75 tissue culture flask at 27°C containing 15 mL of media as described above for 2 days. Cells were harvested by centrifugation at 2500 rpm for 5 min and washed by resuspension in ice cold PBS twice. The washed pellet was then resuspended in five times

its volume of buffer A (15 mM KCL, 10 mM HEPES (pH 7.6), 2 mM MgCl2, 0.1 mM EDTA) and was centrifuged at 5000 rpm at 4°C for 5 min. The cell pellet was then resuspended in the same volume of buffer A supplemented with 1 mM DTT and homogenized with the tight pestle (B) in a Dounce glass homogenizer until most of the cells appeared visually broken-down under the microscope. The cell suspension was then mixed with 1/10 vol of buffer B (1 M KCl, 50 mM HEPES (pH 7.6), 30 mM MgCl2, 0.1 mM EDTA, 1 mM DTT) to increase ionic strength and centrifuged at 10,000 rpm for 10 min at 4°C. The supernatant and pellet were resolved as cytoplasmic and nuclear fractions, respectively.

## Immunoprecipitation (IP)

S2 cells nuclei were isolated as previously described (Khodor et al., 2011), and lysed in 5 vol of lysis buffer (20 mM HEPES pH 7.4, 110 mM potassium acetate, 0.5% Triton X-100, 0.1% Tween-20, 10 mM MnCl2, 1X EDTA-free Complete Protease Inhibitor Cocktail (Roche), 1X PhosStop (Roche) and 50 U/ml Ribolock RNase inhibitor) with 110 U/ml RNase-free DNase (Roche) and incubated for 1 hr at 4°C on rotator. The suspension was centrifuged at 13,000 rpm for 15 min, and the supernatant was transferred in fresh tube and incubated with 20 µL of DynabeadsProtein-G (Thermo Fisher Scientific), which were coated with 5 µg of RNA Pol II Ser2 antibody (Merck Millipore), for 1 hr at 4°C on a rotator. The beads were washed with wash buffer (20 mM HEPES pH 7.4, 110 mM potassium acetate, 0.5% Triton X-100, 0.1% Tween-20, 50 U/mL RNase inhibitor, 4 mM MnCl2) and mixed with cleared cell supernatant and incubated for 1 hr at 4°C on rotator. Beads were washed 3X with wash buffer. For RNase treatment, the beads were incubated with 1 mg/ml RNase in wash buffer, while control sample was incubated in same buffer without RNase at 4°C for 20 min on rotator. The beads were further washed three times with wash buffer and bound proteins were then eluted in 40 µL SDS-sample buffer and boiled 7 min prior loading on the SDS-PAGE gel.

## ChIP-seq and sequence analysis

S2 cells ($10^7$) were fixed in 1% formaldehyde (EM grade, Polyscience) for 15 min at room temperature and then transferred in 125 mM glycine for 5 min to stop the cross-linking reaction. Following centrifugation at 3000 rpm for 5 min at 4°C, the cell pellet was resuspended in 500 µL of PBS containing EDTA-free Complete Protease Inhibitor Cocktail (Roche) and washed twice. The cell pellet was resuspended in 100 µL of SDS lysis buffer (1% SDS, 10 mM EDTA) and sonicated at high intensity for 12 min with 15 s on/off cycles in a Bioruptor sonicator (Diagenode). Samples were diluted in 1 ml ChIP dilution buffer (0.01% SDS, 1.1% Triton X-100, 1.2 mM EDTA, 16.7 mM Tris-HCl pH 8.1, 167 mM NaCl) and centrifuged at 13,000 rpm for 10 min at 18°C. An aliquot (100 µL) of soluble chromatin was kept to extract input DNA. For each ChIP, typically 2 µg of antibody was incubated with 25 µL of DynabeadsProtein-G beads at 4°C overnight. Beads were washed four times with 1 mL of PBS with 5 mg/ml BSA, and resuspended in 40 µL of PBS with 5 mg/mL BSA. The coated beads were then added to the remaining chromatin sample and incubated for two and a half hours at room temperature with rotation. Beads were then sequentially washed with 1 mL of low-salt wash (0.1% SDS, 1% Triton X-100, 2 mM EDTA, 20 mM Tris-HCL pH 8.1, 150 mM NaCl); high-salt wash (as above but with 500 mM NaCl); LiCl wash (0.25M LiCl, 1% IGEPAL-CA630, 1% deoxycholic acid, 1 mM EDTA, 10 mM Tris-HCL (pH 8.1); two 5 min washes in 1 mL of TE buffer (10 mM Tris-HCL (pH 8.0), 1 mM EDTA). The beads were then incubated with elution buffer (0.1M NaHC0$_3$, 1% SDS) at a room temperature for 15 min and reverse crosslinks overnight at 65°C in presence of Proteinase K (2 µL of 50 mg/ml). Chromatin was purified using AMPure XP beads (Beckman Coulter). ChIP and input DNA were further fragmented to 250 bp fragment size using a Covaris S2 focused ultrasonicator prior to library preparation, sequencing and genome mapping was as previously described using SOLiD four genome analyser (Life Technologies) (Kwon et al., 2016). Reads were aligned to the *Drosophila* genome (BDGP R5/dm3 assembly). Average enrichments across gene regions were calculated and plotted prior or after background (input DNA) subtraction. Genes were divided as expressed (RPKM >= 1) or unexpressed (RPKM = 0) based on expression level calculated using *Drosophila* S2 cell expression data (GEO accession no. GSM410195). Average Y14 enrichment was also plotted at expressed gene further divided based on the number of introns (0,1,2,3,4, >= 5) as annotated in R5/dm3 assembly. All our Chip-seq data were deposited in the GEO repository (accession no. GSE84595).

## Polytene chromosomes and whole salivary gland immunostaining

Unless otherwise stated, polytene chromosomes squashes were prepared from salivary glands, dissected in 1X PBS (13 mM NaCl, 0.7 mM $Na_2HPO_4$, 0.3 mM $NaH_2PO_4$, pH 7.4) and transferred to 1% Triton X-100 in 1X PBS for 30 s. Glands were then transferred to 3.7% formaldehyde in 1X PBS followed by 3.7% formaldehyde with 45% acetic acid for 1 min each. Finally, glands were transferred to 45% acetic acid for 1 min and squashed under a coverslip as previously described (*Singh and Lakhotia, 2012*). In some of the experiments, salivary glands were dissected out in a different dissection buffer (15 mM HEPES pH 7.4, 60 mM KCl, 15 mM NaCl, 1.5 mM Spermine, 1.5 mM Spermidine and 1% Triton) and then processed as previously described (*Al-Jubran et al., 2013*; *Rugjee et al., 2013*). Both procedures produce similar immunostaining results. For RNase treatment, salivary glands were dissected in M3 media and incubated in a solution containing 1% Triton X-100 in M3 media for 2 min at room temperature. Glands were then transferred to M3 media with or without 1 mg/ml RNase A (Invitrogen, California, USA) and incubated for 20 min at room temperature. For whole salivary gland immunostaining, larvae were dissected in M3 media, glands were immediately fixed in ice cold PBS with 4% formaldehyde (10% EM grade, Polyscience) and processed as described (*Al-Jubran et al., 2013*). All the glands were mounted in PromoFluor Antifade Reagent (PromoKine). Images were taken using either a Nikon Eclipse Ti epifluorescence microscope, equipped with ORCA-R2 camera (Hamamatsu Photonics) or a Leica TCS SP2-AOBS confocal microscope.

## Polysome profiling and protein extraction

S2 cells were grown in a T75 tissue culture flask as described above until they reached 80% confluence. Cultures were briefly treated with 25 µg/mL cycloheximide for 5 min, cells were harvested by centrifugation at 3000 rpm for 5 min and washed by resuspension in ice cold buffer (10 mM HEPES pH 7.4, 2 mM magnesium chloride, 2 mM magnesium acetate, 100 mM potassium acetate, prepared in DEPC treated $H_2O$). Cells were pelleted again, resuspended and left to lyse by incubating on ice for 10 min in 600 µL lysis buffer (HEPES buffer with 1 mM PMSF, 1 mM DTT, 1.2 µL Ribolock RNase inhibitor, 250 µg/mL heparin, 0.6% Triton X-100 and 1X complete EDTA-free protease inhibitor cocktail). The lysate was cleared twice by centrifugation at maximum speed in a microfuge for 10 min. For RNase-treated samples, RNase inhibitor was excluded from the lysis buffer and 10 µL of 1 mg/mL RNase A was added to the cleared lysate and incubated on ice for 10 min. For EDTA treatment, the cleared lysate was supplemented with 30 mM EDTA for 30 min on ice prior fractionation. For puromycin treatment, 100 µg/mL puromycin was added to the cleared lysate and incubated either on ice or at room temperature for 30 min. The lysis buffer used for the puromycin treatment was as described above, but supplemented with 375 mM KCl and lacked magnesium as previously reported (*Al-Jubran et al., 2013*). The equivalent of about 20 absorbance units at $OD_{260}$ (NanoDrop readings, blanked with water) was loaded onto a 10–50% (w/v) sucrose gradient; gradients were prepared using a gradient mixer which mixed 50% and 10% stock sucrose solutions prepared in polysome buffer (10 mM Tris acetate pH7.4, 70 mM ammonium acetate, 4 mM magnesium acetate, 25 µg/mL cycloheximide in DEPC treated $H_2O$), dispensing directly into SW41 rotor tubes (Beckman Coulter). Puromycin treated samples were loaded on 10–30% sucrose gradients instead of 10–50% prepared as described above but lacking magnesium. Samples were centrifuged at 38,000 rpm for 160 min at 4°C. Fractions were collected by inserting a steel capillary needle to the bottom of the tube, through which the gradient was pumped using a peristaltic pump through an ISCO UA-6 absorbance detector (254 nm) connected to a plotter. Gradients were dispensed as ~800 µL aliquots using a fraction collector. Proteins were extracted using the methanol/chloroform method. A 200 µL aliquot of each fraction was mixed with 800 µL of methanol in a 1.5 mL tube to which 200 µL chloroform was then added and vortexed, followed by addition of 400 µL of $H_2O$. The sample was then vortexed followed by centrifugation at maximum speed for 5 min (proteins/RNA form a white disc at the aqueous/organic interface). The upper phase was discarded and 900 µL of methanol was added, mixed by inversion and centrifuged again for 5 min. The protein-containing pellets were air dried and analyzed by standard SDS-PAGE and Western blotting.

## Acknowledgements

We thank Carl Thummel for providing a clone of the ERE cassette, Ido Golding for plasmids with MS2 repeats, and Julio Vazquez for providing GFP-LacI fly stocks. We also thank Yun Fan and Alicia Hidalgo for fly stocks and experimental advice, as well as Marija Petric for kindly helping with the polysome fractionation and for critically reading the first version of the manuscript. We also thank Bob Michell for comments on parts of the manuscript, and Louise Millward and Vibha Dwivedi for proofreading it. This project was funded by a Leverhulme Trust and Wellcome Trust project grants to SB and BBSRC PhD studentship to TM.

## Additional information

### Funding

| Funder | Grant reference number | Author |
| --- | --- | --- |
| Wellcome | 9340/Z/09/Z | Saverio Brogna |
| Leverhulme Trust | RPG-2014-291 | Saverio Brogna |

The funders had no role in study design, data collection and interpretation, or the decision to submit the work for publication.

### Author contributions

SRC, TM, Acquisition of data, Analysis and interpretation of data, Drafting or revising the article; AKS, Conception and design, Acquisition of data, Analysis and interpretation of data, Drafting or revising the article; MB, Conception and design, Drafting or revising the article, Contributed unpublished essential data or reagents; BJ, PB, Conception and design, Acquisition of data, Contributed unpublished essential data or reagents; AK, Conception and design, Analysis and interpretation of data, Drafting or revising the article; SB, Conceived experiments, Conception and design, Acquisition of data, Analysis and interpretation of data, Drafting or revising the article, Contributed unpublished essential data or reagents

### Author ORCIDs

Saverio Brogna, http://orcid.org/0000-0001-7063-4381

## Additional files

### Supplementary files

• Supplementary file 1. Table of PCR primers used in the study. Sequence of primers for real-time RT-PCR or amplification of dsDNA fragment for in-vitro transcription of dsRNAs used for RNAi. Labels refer to gene names.

### Major datasets

The following dataset was generated:

| Author(s) | Year | Dataset title | Dataset URL | Database, license, and accessibility information |
| --- | --- | --- | --- | --- |
| Brogna S | 2016 | Exon Junction Complex (EJC) proteins bind nascent transcripts independently of pre-mRNA splicing in Drosophila melanogaster | https://www.ncbi.nlm.nih.gov/geo/query/acc.cgi?acc=GSE84595 | Publicly available at the NCBI Gene Expression Omnibus (accession no: GSE84595). |

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
