## [Decision Letter]

Thank you for submitting your article "Exon Junction Complex proteins bind nascent transcripts independently of pre-mRNA splicing in *Drosophila melanogaster*" for consideration by *eLife*. Your article has been favorably evaluated by James Manley (Senior Editor) and three reviewers, one of whom is a member of our Board of Reviewing Editors. The reviewers have opted to remain anonymous.

The reviewers have discussed the reviews with one another and the Reviewing Editor has drafted this decision to help you prepare a revised submission.

Summary:

As you will see, all referees thought that the work was potentially quite interesting and significant but all also raised a number of concerns that must be dealt with before the manuscript can be reconsidered.

1) Although the paper is well done, it is difficult to draw strong conclusions from RNAi-based depletions, as these depletions are not complete and the residual amounts of the proteins may be sufficient to sustain function, at least partially. In particular, for the CWC22 and eIF4AIII depletions, the authors do not show Western blots and thus, the efficiency of the depletions at the protein level has not been validated. Westerns blots should be included for these proteins.

2) Related to CWC22, can the authors rule out that a functional analog operates in *Drosophila*?

3) Although the data supports a model suggesting that Y14 and Mago have functions independent of eIF4AIII and splicing and vice versa, it is possible that EJC deposition occurs at some splice junctions. Essentially the data presented in this manuscript is rather qualitative and lacks sequence resolution to make very strong conclusions. Therefore, the authors should tone down some of the conclusions.

4) The authors should consider the possibility that a majority of these 3 proteins might occur in association with RNA polymerase II and only a small proportion is found in assembled EJCs at steady-state in a cell. The authors would then have observed mainly these pol II-associated proteins in their experiments. Can the authors rule out this interpretation?

The individual reviews are pasted below. Please address the points raised by the reviewers as thoroughly as possible.

*Reviewer #1:*

The manuscript by Brogna and coworkers describes a series of experiments indicating that some core components of the EJC complex associate with nascent transcripts independently of each other, suggesting that the EJC may not assemble in *Drosophila*. In particular the authors investigated the association of eIF4AIII and Y14:Mago heterodimers with polytene chromosomes. They observed that while Y14:Mago colocalize their staining pattern does not always overlap with that of eIF4AIII, although some overlap is observed.

Furthermore the proteins associated with both intron-containing and intronless transcripts suggesting that their binding is not coupled to splicing. Finally, the authors show that depletion of CWC22, which acts downstream of eIF4AIII in human cells, has no effect on eIF4AIII or Y14:Mago deposition, suggesting that models based on vertebrate systems may not apply to invertebrates.

*Reviewer #2:*

Choudhury and colleagues stained polytene chromosome spreads of *Drosophila* salivary glands with anti-EJC factor antibodies and found that eIF4A3 appears to be present at all transcribed loci, whereas Y14 and MAGO co-localized but only were detected at a fraction of the interbands/puffs. Since heat shock loci with and without introns were all stained by all three EJC Abs (eIF4A3, Y14 and MAGO), the authors conclude that the observed Y14 and MAGO pattern does not correlate with the presence of introns in the respective genes. Supporting this conclusion, they also introduced two ecdysone-inducible transgenes, one without and one with introns, into the genome and also reported association of all three EJC factors with both genes (although they only show the data for eIF4A3). A ChIP-seq experiment detected Y14 enriched at transcription start sites, irrespective if the gene had introns or not (Figure 4), but curiously the Y14 association seemed to peak around the transcription start sites (see below).

To analyze the interdependence of these factors' chromatin association, the authors performed knockdowns of CWC22, eIF4A3, Y14 and MAGO in the salivary glands followed by immunostainings of polytene chromosome squashes. CWC22 and eIF4A3 knockdowns gave a negative result, which prevents any firm conclusions. Knockdown of Y14 and MAGOH had no effect on the localization of eIF4A3 but destabilized the respective partner in the heterodimer and hence prevented each other's association with chromatin. Finally, to corroborate their evidence that eIF4A3 associates with RNA independently of Y14/MAGO, the authors showed that eIF4A3 and Y14/MAGO fractionate differently in sucrose gradients, suggesting that the Y14/MAGO heterodimer does not form a stable complex with eIF4A3 on translation mRNA in *Drosophila* cells.

The author's conclusion challenges the current view that *Drosophila* EJC factors form mainly canonical EJCs on mRNA. They propose instead independent association and functions of eIF4A3 and Y14/MAGO. I would however argue that the possibility should be considered that a majority of these 3 proteins might occur in association with RNA polymerase II and only a small proportion is found in assembled EJCs at steady-state in a cell. The authors would then have observed mainly these pol II-associated proteins in their experiments. Can the authors rule out this interpretation?

Specific points to address:

1) It would be nice to see a co-staining with anti-HA and with anti-Y14 or anti-MAGO in Figure 1—figure supplement 3 to confirm that the two Abs give identical staining patterns.

2) Figure 3: Show the Y14 and MAGO stains at the transgene loci along with the eIF4A3 staining. This is a crucial experiment to sustain intron-independent EJC assembly, one of the main conclusions of the paper.

3) Figure 4: How do the authors explain the Y14-enrichment shoulder upstream of the TSS? Assuming random fragmentation of the chromatin, this would indicate that there must be significant amounts of Y14 associated with chromatin upstream of the TSS. Did the authors check in the ChIP experiments if the obtained signal was RNase sensitive? Could it be that the observed enrichment at TSS is due to pol II associated Y14 that is preferentially captured in this ChIP? How long is the average DNA fragment length obtained after the sonication following cell lysis? In my view, the distribution of Y14 in these ChIP-seq experiment does not fit with the authors' overall conclusions.

4) Figure 5: The CWC22 knockdown experiment is inconclusive with respect to the necessity of CWC22 requirement for EJC factor association with chromatin, because it is a negative result and there is a lack of showing how much the CWC22 protein abundance was effectively reduced.

5) Figure 6: For a non-expert, it is difficult to appreciate from the eIF4A3 knockdown glands the interbands in the polytene squash and hence whether Y14 co-localizes there. A co-staining with anti-pol II antibody might makes the picture clearer. In addition, as for CWC22, there is the general issue here that knockdowns yielding negative results (i.e. nothing changes) remain inconclusive.

*Reviewer #3:*

In this manuscript Chouduryet al. study the interaction of Exon Junction Complex (EJC) subunits with the nascent RNA in *Drosophila*. In mammals, the EJC is known to be deposited during splicing at a well-defined position relative to the exon-exon junction and has many roles including earmarking mRNAs containing premature stop codons for degradation. In *Drosophila*, the role of the EJC and its modalities of deposition remain poorly understood, in spite of the conservation of the individual EJC components. The prevailing model for the deposition of the EJC on the RNA posits that the eIF4AIII component is first recruited during splicing via an interaction with a splicing factor, CWC22. In a second time, eIFIIIA recruits the core EJC (MLN51, Y14 and MAGO).

In this study, the authors use a combination of immunostaining and ChIP Seq analyses in *Drosophila* and report the interesting and surprising finding that the recruitment of the core EJC does not require splicing or the CWC22 splicing factor in *Drosophila*. They also show that Y14 and Mago appear to be recruited independently of eIF4AIII to the nascent RNA and have a different distribution in polysomal fraction then the former.

---

## [Author Response]

*Summary:*

*As you will see, all referees thought that the work was potentially quite interesting and significant but all also raised a number of concerns that must be dealt with before the manuscript can be reconsidered.*

*1) Although the paper is well done, it is difficult to draw strong conclusions from RNAi-based depletions, as these depletions are not complete and the residual amounts of the proteins may be sufficient to sustain function, at least partially. In particular, for the CWC22 and eIF4AIII depletions, the authors do not show Western blots and thus, the efficiency of the depletions at the protein level has not been validated. Westerns blots should be included for these proteins.*

There might be some residual CWC22 and eIF4AIII, yet, the no-growth phenotype should indicate that both RNAi knockdowns are efficient in depleting these proteins essential function. As we explained in the Results section, when we drive expression of either RNAi construct early in glands development (with Fkh-Gal4), the salivary glands are underdeveloped to an extent that make it difficult to dissect them and to prepare chromosome squashes (Figure 5). So, to be able do the experiments, we partially circumvent this phenotype by inducing RNAi at a later developmental stage (early 3^rd^ instar larvae). Yet as the larvae are shifted to the RNAi-permissive temperature the salivary glands fail to grow any further, consistent with protein depletion being efficient and rapid. Unfortunately, no antibody is available for fly CWC22, but the mRNA reduction is apparent. It should also be added that dissection of such small glands make it prone to contamination with fragments of fat body (the white-looking tissue seen in Figure 5 which is now labeled with fb on the pictures; these were not targeted by RNAi. Some of the mRNA detected by qRT-PCR is likely to derive from these difficult to avoid contamination even by the most experienced of us. As for eIF4AIII RNAi, even though the glands are particularly small, even under the conditional knockdown setting, immunostaining of the whole glands shows very apparent depletion of the protein in the nucleus and at the chromosomes (Figure 6, panel III and VII respectively). Instead, the signal remains intense in the nuclei of the adhering fatbody. Instigated by the reviewer’s criticism we now dissected enough glands for Western blotting, the result is that also the protein level is very reduced (new Figure 6).

*2) Related to CWC22, can the authors rule out that a functional analog operates in Drosophila?*

There are no apparent sequences in the fly genome which could encode similar proteins, and, given the strong phenotype, genetically it does not seem likely that there is some evolutionarily distant protein that in CWC22 absence could complement its function or functions. We now discuss this point directly in the section where we describe the CWC22 results in the revised manuscript.

*3) Although the data supports a model suggesting that Y14 and Mago have functions independent of eIF4AIII and splicing and vice versa, it is possible that EJC deposition occurs at some splice junctions. Essentially the data presented in this manuscript is rather qualitative and lacks sequence resolution to make very strong conclusions. Therefore, the authors should tone down some of the conclusions.*

We agree and have tried to make it clearer in the revised Discussion (and in the more detailed response to reviewer 3 points) that we cannot formally exclude that EJC deposition occurs at some junctions (such as those reported previously by Ghosh et al. 2012). However, our data indicate that globally these proteins bind nascent RNA independently of splicing and that binding of Y14/MAGO does not require eIF4AIII.

*4) The authors should consider the possibility that a majority of these 3 proteins might occur in association with RNA polymerase II and only a small proportion is found in assembled EJCs at steady-state in a cell. The authors would then have observed mainly these pol II-associated proteins in their experiments. Can the authors rule out this interpretation?*

We cannot exclude that a minor fraction of the proteins associate with Pol II. We had considered this possibility at length. Particularly initially, following some preliminary experiments which indicated that the chromosomal signal was only slightly sensitive to RNase compared to that of Sm proteins (Y12 antibody) which we used as a positive control at the time. However, all subsequent experiments in which we used hnRNPA1 as comparison, showed visibly apparent RNase sensitivity. Perhaps the snRNAs to which Sm proteins are bound are more sensitive to RNase digestion then nascent RNA. Why the EJC signal is not as sensitive to RNase-treatment as that of hnRNPA1, remains unclear. The residual signal might signify some association with Pol II, but it could also mean that these proteins can make secondary contacts with chromatin components or perhaps DNA directly at the loosely packed interbands, which may or may not be of functional significance. It is also possible that such secondary contacts are produced promiscuously during the 20 minutes unphysiological incubation in which gland membranes are permeabilised to allow RNase access. Early in the work, we had done some preliminary Pol II IP in which could not detect any interaction with EJC proteins. In view of the criticism, we have optimized our Pol II solubilization and performed additional Co-IP experiments. The result is that none of the EJC proteins Co-IP with Pol II while general transcription elongation factor Spt6, which was used as positive control clearly does (Figure 4—figure supplement 2). In summary, while we cannot exclude that a minor fraction of the proteins are in contact with Pol II directly, particularly Y14/MAGO as it is not associated to RNA in the cytoplasm, cumulatively the data should indicate that these proteins primarily bind RNA directly at transcription sites, and independently of splicing. We discuss this point further in our response below to reviewer 2 point 3.

*The individual reviews are pasted below. Please address the points raised by the reviewers as thoroughly as possible.*

*Reviewer #2:*

[…]

*The author's conclusion challenges the current view that Drosophila EJC factors form mainly canonical EJCs on mRNA. They propose instead independent association and functions of eIF4A3 and Y14/MAGO. I would however argue that the possibility should be considered that a majority of these 3 proteins might occur in association with RNA polymerase II and only a small proportion is found in assembled EJCs at steady-state in a cell. The authors would then have observed mainly these pol II-associated proteins in their experiments. Can the authors rule out this interpretation?*

See answer to Reviewing Editor point 4 above.

*Specific points to address:*

*1) It would be nice to see a co-staining with anti-HA and with anti-Y14 or anti-MAGO in Figure 1—figure supplement 3 to confirm that the two Abs give identical staining patterns.*

We have not done this staining because the endogenous protein antibody will also detect the tagged protein, therefore such double staining will show “identical” colocalisation, but not informative of whether the signal at the chromosome is from one or the other protein. The colocalisation we have shown between MAGO and HA-Flag double tagged Y14 is unambiguous though (Figure 1—figure supplement 4) and it seems to us a more convincing sign that the tagged proteins are functional, as well as Y14 and MAGO strictly colocalising. That previous staining was with anti-FLAG together with anti-MAGO, in the revision we also show the result of a similar double-staining for Y14-HA and MAGO (new Figure 1—figure supplement 4).

*2) Figure 3: Show the Y14 and MAGO stains at the transgene loci along with the eIF4A3 staining. This is a crucial experiment to sustain intron-independent EJC assembly, one of the main conclusions of the paper.*

We now also show the Y14 staining in Figure 3—figure supplement 1. As we have shown in the Figure 1 and associated supplements, MAGO and Y14 invariably colocalise producing indistinguishable immunostaining patterns. In view of this and the data indicating that the proteins form constitutive heterodimer (Figure 8, see also reviewer 3 point 4 and 5), staining for both proteins seemed unnecessary, this is why here and in later experiments we used only anti-Y14 which is a more sensitive and reliable antibody.

3) Figure 4: How do the authors explain the Y14-enrichment shoulder upstream of the TSS? Assuming random fragmentation of the chromatin, this would indicate that there must be significant amounts of Y14 associated with chromatin upstream of the TSS. Did the authors check in the ChIP experiments if the obtained signal was RNase sensitive? Could it be that the observed enrichment at TSS is due to pol II associated Y14 that is preferentially captured in this ChIP? How long is the average DNA fragment length obtained after the sonication following cell lysis? In my view, the distribution of Y14 in these ChIP-seq experiment does not fit with the authors' overall conclusions.

We don’t yet have an explanation for this interesting “left shoulder”, perhaps some derives from some bidirectional transcription, which though far less prevalent than in mammalian cells it was reported also in *Drosophila* (Core et al., 2012), or, more likely possible, might reflect high occurrence of head-to-head gene pairs in *Drosophila* which are transcribed from separate promoters in opposite directions. An example of such gene pair is shown center of the IGB profile in Figure 4: CG7970-Vta1 which both are expressed in S2 cells (info in Flybase). Clearly these are hypotheses that will need to be addressed directly by future studies.

Unfortunately we have not done the RNase control when the ChIP-seq experiments were originally done due to financial considerations and also in view that such assay is not always straightforward to interpret. Standard ChIP protocols, like the one that it was used here, involves extensive cell fixation which not only crosslinks the protein of interest to the nascent transcript but also indirectly via multiple protein-protein linkages to chromatin. Based on our experience, the apparent RNase sensitivity of the polytene signals is a more convincing indication that the protein associates mostly with the nascent transcript.

We disagree that the ChIP-seq profile does not fit with our overall conclusion. Although, as we discussed above, it is possible that some of Y14 might bind directly to Pol II, it is also feasible that ChIP is biased to detect Y14 that is in close proximity to DNA as this would be less likely to break off during sonication (the sonication cycle we use produces fragments of around 500bp). Additionally, as ChIP and other assays have shown that the majority of Pol II is found near TSS in *Drosophila* as in mammalian cells (Core et al. 2012, and references therein), the prediction would be that a protein associated with the nascent RNA, as short or long as it may be, would produce a TSS enrichment profile in any case, even if there weren’t any bias for ChIP detecting proteins that are tethered closest to DNA. While the mechanism that drives Y14 (and by inference MAGO) to transcription sites, will need to be investigated in future, our ChIP-seq data should demonstrate that beyond reasonable doubts the protein binds active genes independently of splicing. As we discussed above, we could not detect an association with Pol II by IP (see answer to Reviewing Editor point 4).

*4) Figure 5: The CWC22 knockdown experiment is inconclusive with respect to the necessity of CWC22 requirement for EJC factor association with chromatin, because it is a negative result and there is a lack of showing how much the CWC22 protein abundance was effectively reduced.*

Please see response to Reviewing Editor point 1 which was exactly about this.

*5) Figure 6: For a non-expert, it is difficult to appreciate from the eIF4A3 knockdown glands the interbands in the polytene squash and hence whether Y14 co-localizes there. A co-staining with anti-pol II antibody might makes the picture clearer. In addition, as for CWC22, there is the general issue here that knockdowns yielding negative results (i.e. nothing changes) remain inconclusive.*

Although the chromosomes are very underdeveloped in these minute glands, the characteristic pattern of intensely stained DAPI bands flanked by weakly stained interbands is visible. This should now be more apparent in the new figure in which we show, as suggested, Pol II and Y14 double staining (Figure 6—figure supplement 1). While the stainings banding pattern is not as apparent as in wildtype chromosomes, there are segments where a banding pattern is visible, and at these, co-localization of Y14 with Pol II is evident (panel V). Additionally, as on wildtype chromosomes (Figure 1—figure supplement 2), there are many sites which show high Pol II but weak Y14 signal.

As for the criticism of the CWC22 RNAi experiment, please see above answer to the Reviewing Editor point 1.